# A transfer-learning approach to predict antigen immunogenicity and T-cell receptor specificity

Barbara Bravi[1,2]*, Andrea Di Gioacchino[2†], Jorge Fernandez-de-Cossio-Diaz[2†], Aleksandra M Walczak[2‡], Thierry Mora[2‡], Simona Cocco[2‡], Rémi Monasson[2‡]

[1]Department of Mathematics, Imperial College London, London, United Kingdom; [2]Laboratoire de Physique de l'Ecole Normale Supérieure, ENS, Université PSL, CNRS, Sorbonne Université, Université Paris-Cité, Paris, France

**Abstract** Antigen immunogenicity and the specificity of binding of T-cell receptors to antigens are key properties underlying effective immune responses. Here we propose diffRBM, an approach based on transfer learning and Restricted Boltzmann Machines, to build sequence-based predictive models of these properties. DiffRBM is designed to learn the distinctive patterns in amino-acid composition that, on the one hand, underlie the antigen's probability of triggering a response, and on the other hand the T-cell receptor's ability to bind to a given antigen. We show that the patterns learnt by diffRBM allow us to predict putative contact sites of the antigen-receptor complex. We also discriminate immunogenic and non-immunogenic antigens, antigen-specific and generic receptors, reaching performances that compare favorably to existing sequence-based predictors of antigen immunogenicity and T-cell receptor specificity.

*For correspondence:
b.bravi21@imperial.ac.uk

†These authors contributed
equally to this work
‡These authors also contributed
equally to this work

## Editor's evaluation

In this important work, the authors present a sequence-based approach using transfer learning and Restricted Boltzmann Machines to predict antigen immunogenicity and specificity. The evidence and methodology are compelling. This work should be of interest to immunologists, computational biologists, and biophysicists.

## Introduction

T cells play an essential role in the immune response to pathogens and malignancies. Killer T cells are activated following the binding of their surface receptors (T-cell receptors or TCRs) to short portions of pathogen-related proteins (peptide antigens) that are presented by class I major histocompatibility complexes (MHCs) forming the peptide-MHC epitope (pMHC).

Only a fraction of peptides presented by the MHC are immunogenic, meaning that they possess biochemical properties that can promote a T-cell response (*Sette et al., 1994*). Accurate prediction of immunogenicity is crucial to the successful identification of microbial antigens and cancer neoantigens (antigens carrying cancer-related mutations) that help develop vaccines and immune-based cancer therapies. A very recent systematic assessment *Buckley et al., 2022* of the available models to identify immunogenic targets from pathogens and cancers reports suboptimal overall performances, with none of the models able to substantially improve beyond pure MHC-presentation prediction when evaluated on immunogenic peptides from a new virus (SARS-CoV-2). The largest-scale validation *Wells et al., 2020* of existing computational pipelines for neoantigen discovery has highlighted

a general lack of consensus among their predictions and a rather low average success rate, with only 6% of the predicted neoantigens validated as truly immunogenic.

We also know that a given pMHC epitope elicits the response of only specific small subsets of the human T-cell repertoire. Predicting the molecular composition of TCRs that have the potential to be reactive to a given epitope is a difficult computational problem that is yet not fully solved despite numerous recent advances (*Gielis et al., 2019*; *Jokinen et al., 2019*; *Springer et al., 2020*; *Montemurro et al., 2021*; *Weber et al., 2021*). Improvements in predicting immunogenicity and TCR specificity would have direct consequences for medical applications, including the study of an individual's infection history from their T-cell repertoire and personalized adoptive T-cell therapy for cancer treatment.

A few approaches have been proposed to predict antigen immunogenicity (*Calis et al., 2013*; *Trolle and Nielsen, 2014*; *Chowell et al., 2015*; *Ogishi and Yotsuyanagi, 2019*; *Gao et al., 2020*; *Schmidt et al., 2021*; *Lin et al., 2021*; *Buckley et al., 2022*), some of them specifically developed for computational pipelines of neoantigen discovery (*Łuksza et al., 2017*; *Smith et al., 2019*; *Riley et al., 2019*; *Schenck et al., 2019*; *Schaap-Johansen et al., 2021*). The model by *Calis et al., 2013* is currently the most used resource for MHC class I immunogenicity prediction, being both implemented by the IEDB tool (http://tools.iedb.org/immunogenicity/) for immunogenicity predictions and integrated in the T cell-antigen interaction prediction by the NetTepi server (*Trolle and Nielsen, 2014*). It relies on measuring the enrichment in amino-acid usage between immunogenic and non-immunogenic peptides, assuming position independence of the main biophysical properties underlying immunogenicity. Some of these approaches are based on a preliminary choice of peptide positions (*Schmidt et al., 2021*) or properties (such as hydrophobicity *Chowell et al., 2015*), assumed to be important for recognition by TCRs.

Conversely, the computational prediction of the epitope specificity of a given TCR is extremely relevant for therapeutic design, and has been done so far through a variety of machine-learning methods (*Cinelli et al., 2017*; *Gielis et al., 2019*; *Jokinen et al., 2019*; *Davidsen et al., 2019*; *Springer et al., 2020*; *Luu et al., 2021*; *Sidhom et al., 2021*; *Chronister et al., 2021*; *Weber et al., 2021*; *Lin et al., 2021*; *Montemurro et al., 2021*; *Zhang et al., 2021*). These models are typically calibrated towards achieving high predictive power as classifiers of TCR specificity, while less attention is paid to the interpretability of their predictions in terms of molecular properties determining binding specificity, apart from a few exceptions relying on model-agnostic interpretability pipelines (*Papadopoulou et al., 2022*).

Both antigen immunogenicity and epitope-specificity of T-cell receptors have a molecular-level component. They result from specific physico-chemical constraints on the sequence composition of antigens and T-cell receptors. Immunogenic antigens and epitope-specific receptors display an enrichment in specific patterns of amino acid composition. For example, several works have shown enrichment in hydrophobic (*Chowell et al., 2015*; *Riley et al., 2019*) and aromatic (*Schmidt et al., 2021*) residues in immunogenic peptides, compared to all presented peptides (which are predominantly non-binders of TCRs). TCRs specifically responding to the same peptide are also characterized by convergent amino-acid motifs (*Cinelli et al., 2017*; *Dash et al., 2017*; *Glanville et al., 2017*; *Pogorelyy et al., 2019a*, *Huang et al., 2020*; *Shomuradova et al., 2020*; *Mayer-Blackwell et al., 2021*; *Minervina et al., 2021*; *Goncharov et al., 2022*), whose retrieval is the focus of several clustering approaches (*Dash et al., 2017*; *Glanville et al., 2017*; *Meysman et al., 2019*; *Pogorelyy and Shugay, 2019b*; *Thakkar and Bailey-Kellogg, 2019*; *Mayer-Blackwell et al., 2021*; *Valkiers et al., 2021*). Such specific patterns are observed in addition to others, broadly shared across antigens and TCRs. These shared patterns reflect baseline constraints ensuring viability and function (ensuring that TCRs are structurally stable and have the basic binding properties allowing them to pass thymic selection, or ensuring that antigens have high binding affinity to the presenting HLA protein). An outstanding question is how to disentangle sequence pattern enrichment underlying immunogenicity and TCR epitope specificity from the stronger statistical signatures stemming from these baseline constraints. This separation could generate insight into the molecular basis of antigen immunogenicity and epitope specificity and could enable their prediction from sequence alone.

To tackle this question, we here introduce a strategy of 'differential learning' within the architecture of Restricted Boltzmann Machines (*Hinton, 2002*; *Hinton and Salakhutdinov, 2006*), which we call diffRBM (differential Restricted Boltzmann Machine). DiffRBM relies on a transfer learning procedure,

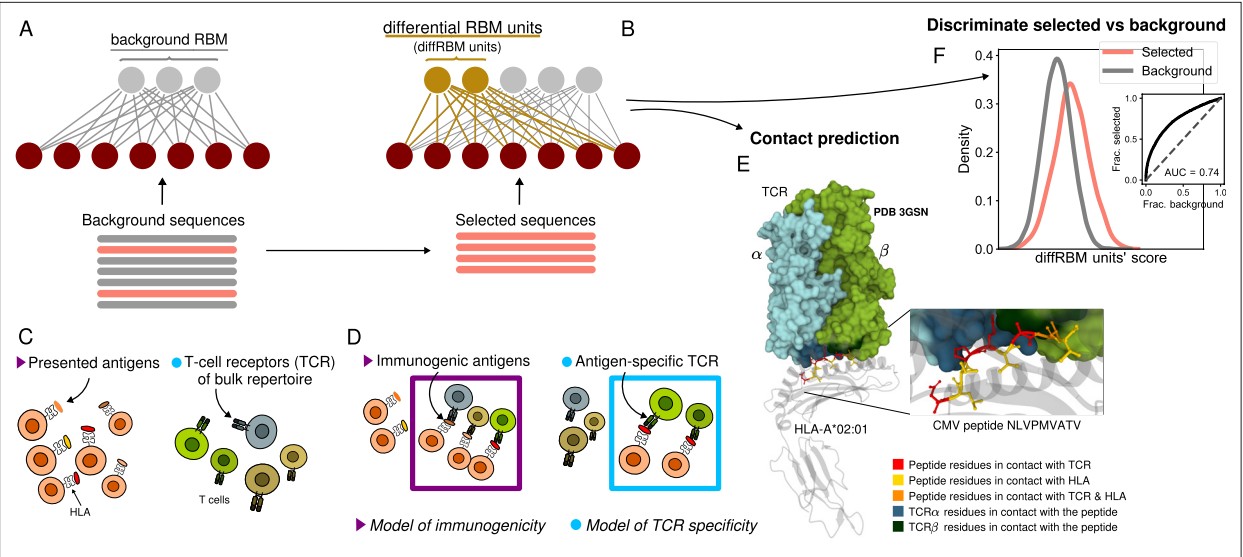

**Figure 1.** Cartoon of the differential RBM (diffRBM) learning approach. (**A**) The parameters of background RBM (gray) are learnt from the 'background' sequence dataset. (**B**) The diffRBM units (gold) are learnt from a small subset of 'selected' sequences. (**C**) We consider the application of diffRBM to modeling peptide immunogenicity or T-cell receptor (TCR) antigen specificity, whereby the background dataset consists, respectively, of all antigens presented by a given Human Leukocyte Antigen class I complex (HLA) or of generic TCRs from the bulk repertoire. (**D**) The selected sequences correspond to HLA-specific antigens validated to be immunogenic or to TCRs that are antigen-specific responders. The inferred parameters associated to the diffRBM units allow one to identify putative contact positions in the peptide-HLA-TCR structure (**E**) and more generally to assign scores that distinguish the selected from the background sequences (**F**). E is an example of a peptide-HLA-TCR structure for the CMV peptide NLVPMVATV (PDB-ID:3GSN), where the contact points along the peptide and the TCR are highlighted in different colors (image obtained with Mol* *Sehnal et al., 2018*).

The online version of this article includes the following figure supplement(s) for figure 1:

**Figure supplement 1.** DiffRBM architecture recapitulating the mathematical notation used in Materials and methods.

where we first learn general background distributions of antigen or TCR sequences, exploiting the large availability of such data. We then refine these models to learn, typically from small amounts of sequence data, the distinctive features that confer immunogenicity or epitope specificity. We first inspect and extract biologically interpretable features from the trained parameters. We then assess diffRBM performance at predicting antigen immunogenicity and TCR specificity compared to existing computational tools.

## Overview of diffRBM

In this paper, we develop the diffRBM approach to build models of peptide immunogenicity and T-cell binding to specific peptides. The basic transfer-learning idea of diffRBM is in principle applicable to any data that have some distinctive features compared to a much larger pool of data endowed with the baseline properties. This is the reason why we will refer to these two different sets of data generically as 'selected' and 'background' datasets (*Figure 1A–B*). Here, these two sets correspond to, respectively, immunogenic and presented only antigens in the case of the model of immunogenicity, or antigen-specific and bulk-repertoire TCRs in the case of the model of TCR epitope specificity (*Figure 1C–D*).

The portion of the machine learnt from the background dataset specifies what we call 'background RBM' (*Figure 1A*), while we call 'differential' the hidden units learnt from the selected dataset, since they focus the learning on its distinctive features (*Figure 1B*). We will call diffRBM (standing for 'differential RBM') the full model architecture comprising background RBM and the diffRBM units (*Figure 1B*, *Figure 1—figure supplement 1*).

Our differential approach is akin to the machine-learning technique known as transfer learning, whereby a model learnt for one task is transferred to the second task in such a way that the information embedded in the first model facilitates the learning of the second model. Deep learning approaches like (*Wu et al., 2021*; *Akbar et al., 2022*; *Leem et al., 2022*) pre-train on large sets of unannotated antibody sequences to learn the general structure of the space before fine tuning the parameters

on sequences with a defined epitope specificity. Similarly, transfer-learning approaches aimed at the prediction of TCR-antigen binding take advantage of pre-training on larger amounts of data, either describing general protein-protein interactions (*Weber et al., 2021*) or sets of TCRs and epitopes without a specific pairing (*Lu et al., 2021*). The SONIA method (*Elhanati et al., 2014*; *Sethna et al., 2020*) (and its deep-learning version soNNia *Isacchini et al., 2021*) quantifies thymic selection on top of the distribution of receptor generation via V(D)J recombination. Contrary to SONIA, diffRBM predicts also antigen immunogenicity. The diffRBM units focus on the differences relative to the background, disentangling in this way the data features that make them a selected subset with distinctive properties. As a result, scores based on the diffRBM units distinguish the selected data from the background data (*Figure 1F*), performing better than RBM models learnt without a transfer-learning step. As a consequence, diffRBM units and their parameters support the discovery of salient amino acid patterns underlying TCR-antigen binding, for instance they can identify antigen-TCR contact points in the three-dimensional molecular structure (*Figure 1E*).

## Results

### DiffRBM model of antigen immunogenicity

We collected from the Immune Epitope Database (IEDB; *Vita et al., 2019*) sets of peptides that elicited a T-cell reaction in T cell assays (referred to as 'immunogenic') and sets of peptides that were not T-cell-reactive ('non-immunogenic'). The peptide-presenting MHCs are specialized proteins coded by highly polymorphic human genes called Human Leukocyte Antigen (HLA) gene. We selected only peptides presented by 3 HLA-I alleles (HLA-A*02:01, HLA-B*35:01, HLA-B*07:02). We chose these HLA-I alleles since they are associated to at least 200 immunogenic peptides in IEDB and at least one TCR-pMHC structure in the Protein Data Bank (Materials and methods). We trained diffRBM for each set of HLA-restricted immunogenic peptides in two steps, by training first a background RBM on samples of antigens presented by one specific HLA via the RBM-based algorithm RBM-MHC (*Bravi et al., 2021b*), and next by training the diffRBM units on the set of immunogenic antigens of the same HLA type (*Figure 2A*, *Figure 2—figure supplement 1*). Background RBM can predict scores of presentation on the specific HLA under consideration, while the diffRBM units predict scores of immunogenicity.

Differently from some existing approaches to modeling immunogenicity (*Calis et al., 2013*; *Chowell et al., 2015*; *Schmidt et al., 2021*), we train *HLA-specific* models. Preliminary inspection of the datasets revealed that patterns of amino acid enrichment differ between immunogenic peptides presented by different HLAs, apart from some general trend in terms of dominant amino acid properties (*Calis et al., 2013*; *Chowell et al., 2015*; *Riley et al., 2019*; *Schmidt et al., 2021*; *Łuksza et al., 2022*). This is true also when we restrict to the peptide positions known to be relevant for immunogenicity (*Schmidt et al., 2021*; *Rudolph et al., 2006*; *Calis et al., 2012*; *Łuksza et al., 2022*; *Figure 3A*). For instance, the extent to which enrichment in hydrophobicity can discriminate immunogenic from non-immunogenic peptides was observed to vary across HLAs (*Buckley et al., 2022*), supporting HLA-specific strategies to model immunogenicity. On the practical side, cross-HLA imbalances in the size of training sets were found to skew predictions toward the most characterized HLAs, in particular toward HLA-A*02:01 (*Buckley et al., 2022*).

Equipped with these single-alleles models of immunogenicity, we can perform two tasks, the prediction of peptide sites in contact with the TCR (*Figure 1E*) and the classification of immunogenic peptides against non-immunogenic ones (*Figure 1F*).

### Validation of model predictions against TCR-pMHC structures

Since the diffRBM units are inferred to capture the distinctive patterns of peptide immunogenicity, we hypothesized that the associated inferred parameters can be informative about the actual structural properties of the pMHC-TCR complex. To test this hypothesis, we collected a set of resolved crystal structures publicly available in the Protein Data Bank (*Berman et al., 2000*) describing peptides in complex with the presenting HLA molecule and a cognate TCR. For each of these TCR-pMHC complexes, we estimated the peptide positions in contact with the TCR and the HLA (Methods, *Figure 2—source data 1*). *Figure 2B–C* show the frequency of contacts at each peptide position with the HLA (B) and the TCR (C) in peptides presented by HLA-A*02:01 (the HLA allele to which

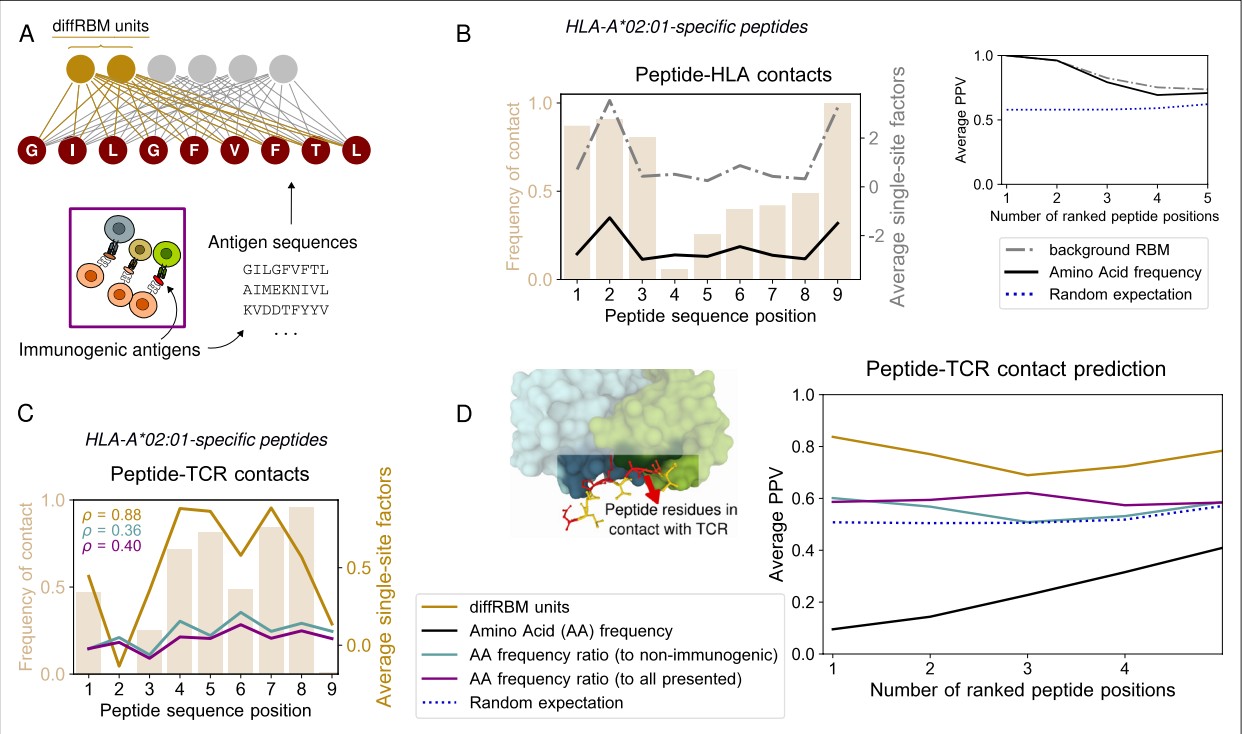

**Figure 2.** DiffRBM model of immunogenicity and structural interpretation of its parameters. (**A**) DiffRBM units are learnt from HLA-specific peptides annotated as immunogenic. (**B**) HLA contact frequency for each peptide position across 41 structures (bars, left-axis). On the right-axis, log-frequency of amino-acids in the background dataset of HLA-A*02:01-presented antigens (black line), and single-site factor magnitude predicted by the background RBM (HLA-A*02:01-specific presentation model, gray line), both averaged over the 41 structures. Right inset: Average Positive Predictive Value (PPV) for the prediction of peptide positions in contact with the HLA as a function of the number of ranked positions, averaged over the 41 structures. The average PPV over a uniformly random prediction is shown in blue (dotted line, see Materials and methods). (**C**) Same as B, but for peptide-TCR contacts. Single-site factors as calculated from the diffRBM units of the immunogenicity model. Immunogenic to either non-immunogenic or all presented peptides' amino acid frequency ratios are also shown (legend in D). $\rho$ denotes the correlation coefficient between the contact frequency distribution and single-site factor magnitudes. Peptide contact positions are those within 3.5 Å (4 Å) to the HLA (TCR) in the crystal structure. (**D**) Peptide-TCR contact prediction PPV for each peptide position, sorted by single-site factor magnitude, and averaged over 46 structures (4 for HLA-B*35:01, 41 for HLA-A*02:01, 1 for HLA-B*07:02). Predictions are made using the HLA-specific immunogenicity model for each peptide. Average PPVs are reweighed by a sequence similarity between peptide entries, see Materials and methods (*Figure 2—figure supplement 3A–B*, *Figure 2—figure supplement 4A–B*).

The online version of this article includes the following source data and figure supplement(s) for figure 2:

**Source data 1.** List of TCR-pMHC structures from PDB and estimated contact positions at 4Å.

**Figure supplement 1.** Schematic summary of the construction of a diffRBM model of immunogenicity.

**Figure supplement 2.** Hyperparametric search for the diffRBM model of immunogenicity.

**Figure supplement 3.** Prediction of peptide contact positions with the TCR.

**Figure supplement 4.** Prediction of peptide contact positions with the HLA.

the large majority of structures is available, 41 over 46 structures, see Materials and methods). These contact frequency distributions highlight that positions 2 and 9 (and 1 and 3 to a lower extent) are the anchor sites for the binding of the peptide to the HLA-I protein (*Figure 2B*), while central positions (4-8) tend to be in contact to the TCR (*Figure 2C*), consistently with the analyses of structures reported in *Rudolph et al., 2006*; *Calis et al., 2012*; *Schmidt et al., 2021*; *Milighetti et al., 2021*. Previous measures of TCR functional avidity with mutant peptides indicate that amino acid changes at the peptide central positions impact the most T-cell activation (*Hoof et al., 2010*; *Schmidt et al., 2021*; *Łuksza et al., 2022*), suggesting that these positions are important for TCR response. Anchor sites of peptide-HLA binding can be inferred from the background RBM parameters. We first focus on HLA-peptide binding. Anchor sites for the bond with the HLA constrain the amino acid usage at those positions across peptides presented by that same HLA, increasing the frequency of the amino acids required for binding (*e.g.* I, L, V at positions 2 and 9 of HLA-A*02:01 ligands). As a result,

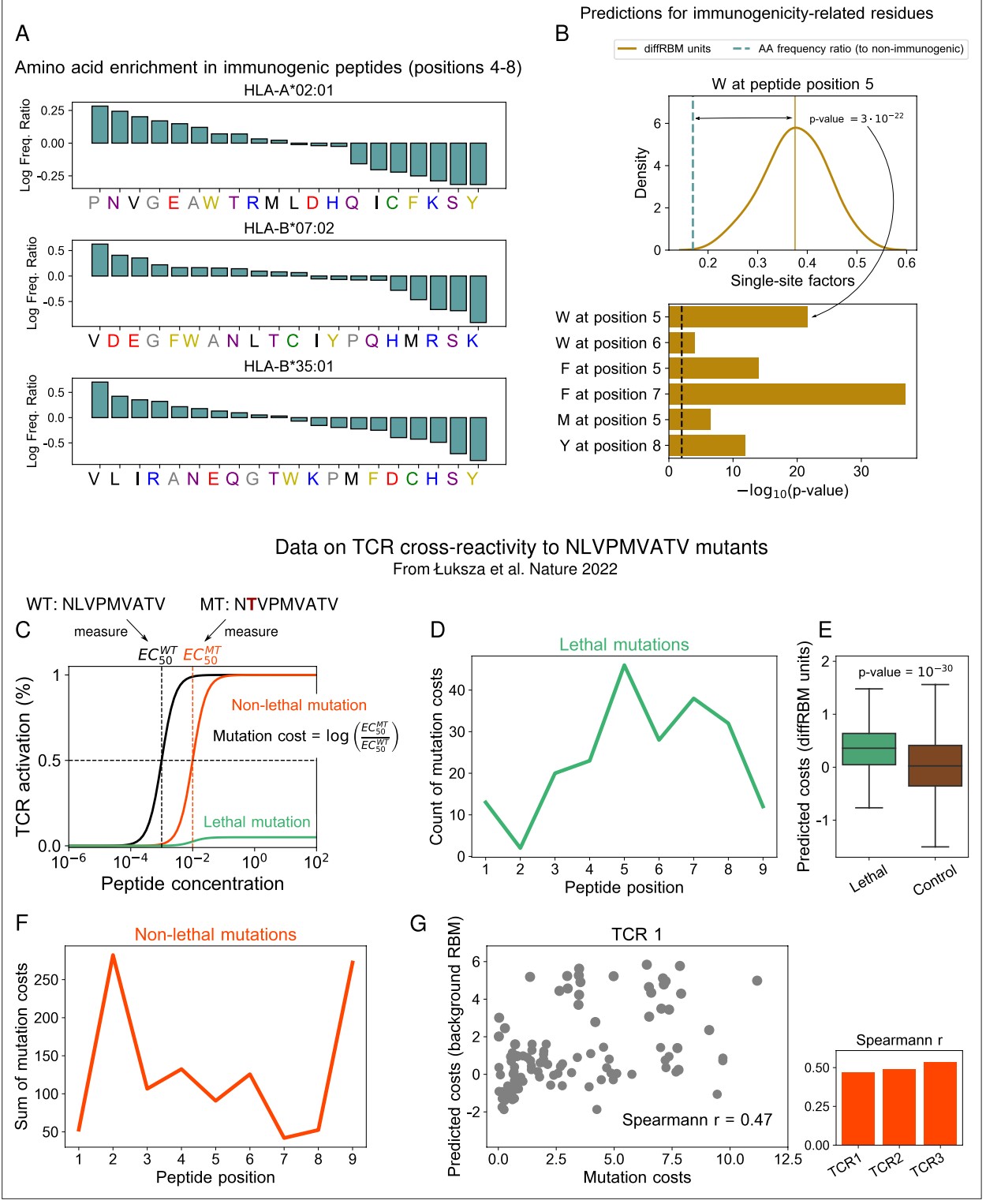

**Figure 3.** DiffRBM units encode molecular features of immunogenicity. (**A**) Amino-acid usage log-enrichment of immunogenic to non-immunogenic peptides, across central positions (4-8) for each HLA type. The color code indicates amino acid properties: negatively charged (red), positively charged (blue), polar (purple), aromatic (yellow), aliphatic hydrophobic (black), cysteine (green), tiny (grey). (**B**) DiffRBM predicts a positive contribution to immunogenicity of key residues, in agreement with observations. (Top) DiffRBM single-site factors distribution evaluated across HLA-A*02:01-specific immunogenic sequences with W at position 5. (Bottom) The single-site factors given by the immunogenic *vs* non-immunogenic amino acid frequency ratio, which do not include the sequence context (Materials and methods), predict a much lower contribution to immunogenicity, as indicated by the p-values of their difference with respect to the average of the diffRBM single-site factors distribution. (**C**) Illustration of TCR activation curves from

*Figure 3 continued on next page*

*Figure 3 continued*

*Łuksza et al., 2022* for wild-type (*WT*) peptide NLVPMVATV and its point-mutants (*MT*). (**D**) Total count of lethal mutation costs (214 of 513 TCR-mutant combinations), plotted per mutated peptide position. (**E**) DiffRBM units predicted costs of lethal mutations are mostly positive (Materials and methods). (**F**) Non-lethal mutation costs sum (299 of 514 TCR-mutant combinations) per mutated peptide position. (**G**) Experimental *vs* background RBM predicted costs for non-lethal mutations, for one TCR (TCR1). Spearmann correlation coefficients *r* are comparable across all 3 TCRs, with p-values ≤10⁻⁶ (*Figure 3—figure supplement 1B*).

The online version of this article includes the following figure supplement(s) for figure 3:

**Figure supplement 1.** Prediction of immunogenicity-related residues and mutation costs.

the sites occupied by the high-frequency amino acids at those positions are typically anchor sites (*Figure 2B*). To check whether the background RBM captures this statistical information, we took the parameters learnt to reproduce the amino-acid statistics in HLA-A*02:01-presented peptides and we used them to define peptide site-specific quantities that we call 'single-site factors'. These correspond essentially to the background RBM log probabilities of a given residue (Materials and methods). For each peptide found in the TCR-pMHC-A*02:01 structures, we ranked sequence positions by single-site factors and we verified this prediction against the true contacts by calculating an average Positive Predictive Value (PPV), see the inset of *Figure 2B* and Materials and methods. Large single-site factors (dashed-dotted black line), similarly to the amino acid frequency, detect the peptide-HLA binding anchor sites, both when the rest of the sequence is accounted for and when it is not (the average PPV of both these predictions is comparable to the amino acid frequency-based prediction, see *Figure 2B* and *Figure 2—figure supplement 4A*). DiffRBM single-site factors flag up peptide positions important for immunogenicity. Also in sets of immunogenic peptides we expect the statistics at the contact positions with the TCR to reflect the constraint of being in contact, when compared to the statistics of all presented peptides. These constraints are captured by the parameters linked to the diffRBM units. In analogy to the prediction of HLA-peptide binding via background RBM, we introduce 'diffRBM single-site factors' to predict the single residue potential to establish a contact with the TCR. The diffRBM single-site factors give approximately log odds-ratios between the full RBM and the background RBM probabilities of a certain residue conditional on the rest of the sequence (*Equation 11* in Materials and methods). Once evaluated on an amino acid in a given peptide position, these model-dependent terms provide a measure of the predicted contribution to immunogenicity of the amino acid, while accounting for the sequence context given by all other sites. *Figure 2C* shows the average magnitude of the diffRBM single-site factors evaluated on the 41 peptides binding to HLA-A*02:01, which identifies positions 4–8 as the most relevant for immunogenicity. The pattern of positional contact frequency with the TCR in the same figure supports the structural interpretation of the model's prediction in terms of binding between peptide and TCR. The diffRBM units' prediction recovers the pattern of positions important for immunogenicity without restricting a priori the input sequences to a subset of peptide positions already known or assumed to be involved in TCR binding, in contrast with existing approaches (*Calis et al., 2013*; *Schmidt et al., 2021*) that choose a priori what positions to retain in the formulation of the immunogenicity model. DiffRBM single-site factors predict peptide contact positions with the TCR. We ranked sequence positions by the diffRBM single-site factors' magnitude for each peptide in the TCR-pMHC complexes, and we took the highest ranking positions as predicted contact points. The peptide-averaged PPV for this prediction as a function of the ranked positions (*Figure 2D*) indicates a model's predictive power substantially higher than the random expectation (the p-value of this difference at the first ranked position is $7.7 \times 10^{-5}$, see Materials and methods). We compared the prediction by the diffRBM units to predictions based on the enrichment in amino acid usage in immunogenic peptides (*Figure 2C–D*). In this case, we ranked positions based on the log ratio between the position-specific amino acid frequency in immunogenic peptides of a given HLA type and the one in the set of either all presented peptides or the non-immunogenic peptides with the same HLA type (Materials and methods). The diffRBM units outperform these predictions based on amino acid frequency ratios, as quantified by the average PPV (*Figure 2D*, *Figure 2—figure supplement 3*). For the HLA-A*02:01 peptides, we also found that the magnitude of the diffRBM single-site factors across positions correlates with the pattern of contact frequency better than predictions based on amino acid frequency ratios (*Figure 2C*).

## DiffRBM encodes molecular features of immunogenicity

Next, we assessed whether our model predictions of the residues' contribution to immunogenicity, based on single-site factors, are consistent with previous findings (*Piepenbrink et al., 2013*; *Chowell et al., 2015*; *Riley et al., 2019*; *Schmidt et al., 2021*; *Łuksza et al., 2022*). We inspected the amino acid enrichment with respect to non-immunogenic antigens at the common peptide-TCR contact points (positions 4–8) in *Figure 3A*. Some of the observed trends resemble the ones found in previous studies, for instance: the bias, in immunogenic sequences, towards hydrophobic amino acids (*Chowell et al., 2015*; *Riley et al., 2019*; *Łuksza et al., 2022*; *Buckley et al., 2022*) (especially valine, V); the abundance of glutamic acid (E) (*Calis et al., 2013*) and, to a more moderate extent, of tryptophan (W) *Schmidt et al., 2021*; *Łuksza et al., 2022*; the depletion of the small polar amino acid serine (S) and the positively charged amino acid lysine (K), consistently with the observations respectively in *Chowell et al., 2015* and *Schmidt et al., 2021*. We noted also a number of discrepancies, presumably due to the use of updated datasets of immunogenic and non-immunogenic peptides and the restriction to sequences of only three selected HLA types. The most striking is the under-representation of the aromatic amino acids phenylalanine (F) and tyrosine (Y), particularly severe in HLA-A*02:01, in contrast with the experiment-based observations in *Schmidt et al., 2021*; *Piepenbrink et al., 2013*. Note however that also other analyses performed on IEDB data, like *Calis et al., 2013*; *Chowell et al., 2015*, did not flag up a significant enrichment in Y. DiffRBM single-site factors recover the positive contribution to immunogenicity of key residues. We considered a few combinations of amino acids and positions along the peptide that were suggested to play a crucial role in T-cell reactivity and binding (in the context of HLA-A*02:01 epitopes) based on structural or functional analyses. For instance, W at position 6 and F at position 7 (*Schmidt et al., 2021*), as well as Y at position 8 (*Piepenbrink et al., 2013*), were observed to form a variety of stabilizing interactions with the TCR. Testing functional avidity of TCRs against peptides harbouring single-point mutations, (*Schmidt et al., 2021*) detected that F and W at position 5 triggered the strongest activation signal, while *Łuksza et al., 2022* found that the substitution of methionine (M) at position 5 systematically abrogated TCR response.

For each of these combinations, we identified the sequences possessing the particular amino acid at the combined position and calculated the diffRBM single-site factors for that position (*Figure 3B*). For all amino acid/position pairs, the distribution of these single-site factors is skewed toward positive values (*Figure 3B*, *Figure 3—figure supplement 1A*), meaning that the diffRBM units predict a positive contribution to immunogenicity. In contrast, measuring purely the amino acid frequency ratio at that position under the independent-site assumption predicts a contribution to immunogenicity that is significantly smaller (*Figure 3B*). In other words, the diffRBM's ability to capture the sequence context reconciles its predictions with previous findings on residues that are key to immunogenicity.

Model predictions are in agreement with data on TCR reactivity to mutant peptides. To further corroborate the connection between the model's predictions and T-cell reactivity assays, we considered the data from *Łuksza et al., 2022* on the TCR response to one of the highly immunogenic peptides to which our HLA-A*02:01-specific diffRBM model of immunogenicity can be applied (NLVPMVATV from the human cytomegalovirus). These data measure TCR reactivity to all possible single-site mutants of NLVPMVATV for 3 TCRs specific to it (Materials and methods, *Figure 3C*). Some of these single-site mutations do not cause a complete loss of TCR response ('non-lethal' mutations), meaning that TCR reactivity can be recovered by increasing the peptide concentration (*Figure 3C*). We can estimate a 'cost' for such mutations in terms of the TCR cross-reactivity between NLVPMVATV and its mutants, measured in *Łuksza et al., 2022* as the log ratio between the half maximal effective concentration for TCR activation after the peptide has been mutated ($EC_{50}^{MT}$) and the one before the mutation ($EC_{50}^{WT}$), see *Figure 3C*. Other mutations completely destroy peptide-TCR binding, and TCR reactivity cannot be restored even at highest peptide concentrations considered ('lethal' mutations, see *Figure 3C*).

Lethal mutations tend to occur at the typical TCR-contact positions along the peptide (*Figure 3D*), and we confirmed that the diffRBM single-site factors predict mainly positive costs for these mutations (see Materials and methods), which matches qualitatively the observed loss of immunogenicity (*Figure 3E*).

We observed that the magnitude of non-lethal mutations across all TCR-mutant pairs is concentrated at the peptide positions 2 and 9, which are the anchor sites for binding the HLA in HLA-A*02:01-specific peptides (*Figure 3F*). We therefore hypothesized that these mutation costs, despite

being measured in terms of loss of TCR reactivity, could reflect a disruption of the peptide presentation on the HLA. We found that the model prediction for HLA-A*02:01 presentation (background RBM) correlates in a statistically significant way to the experimental mutation costs, and the degree of correlation is consistent across all three TCRs (*Figure 3G*, *Figure 3—figure supplement 1B*). In contrast, the diffRBM single-site factors cannot predict these mutational effects concentrated at the anchor sites for presentation (*Figure 3—figure supplement 1B*), since its parameters capture the distinctive molecular composition of immunogenic peptides at the central positions (*Figure 2C*).

## DiffRBM discriminates immunogenic *vs* non-immunogenic peptides

The diffRBM units learn distinctive sequence patterns of immunogenicity, having the background RBM captured the sequence constraints associated to presentability. Such patterns should contribute to

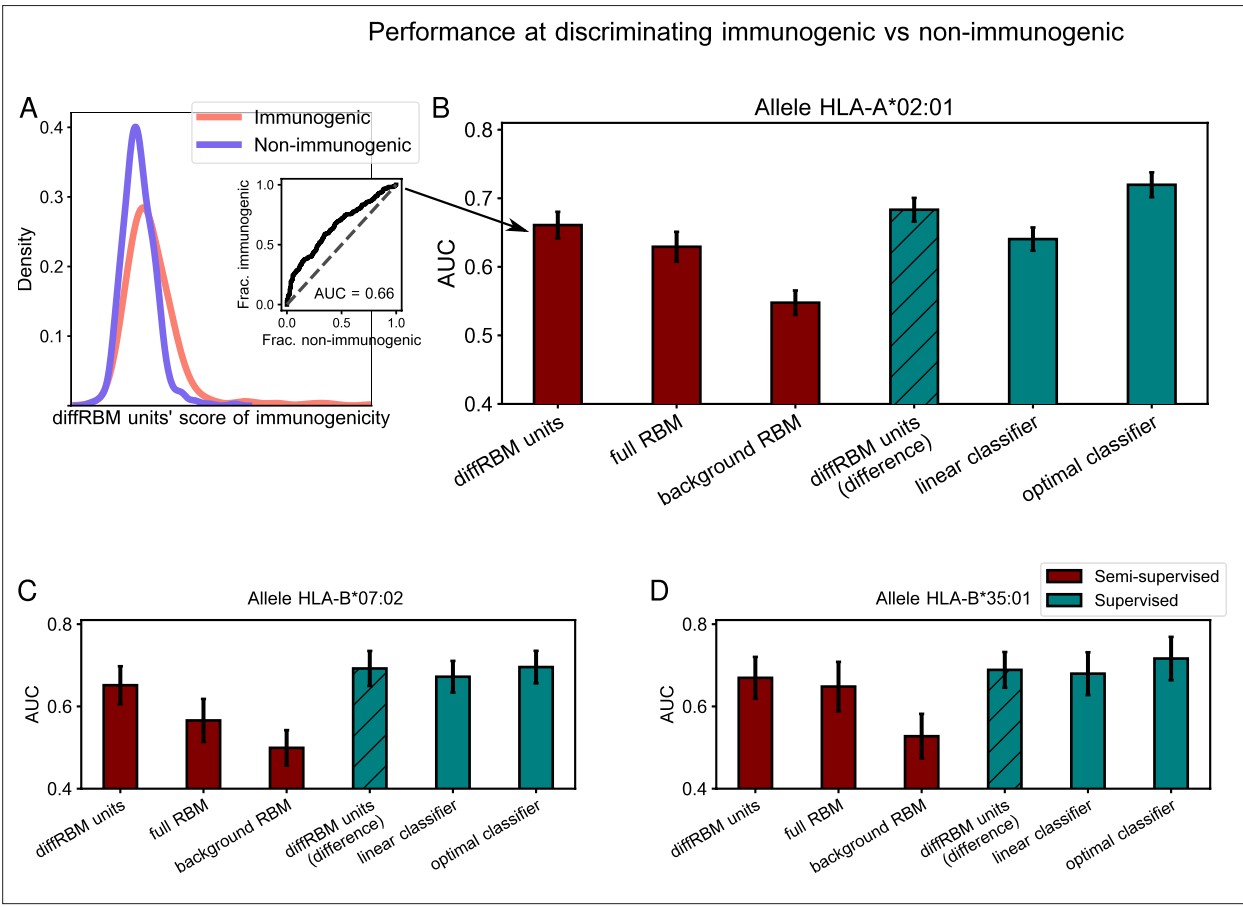

**Figure 4.** Immunogenic *vs* non-immunogenic peptide discrimination performance. (**A**) The Area Under the Curve (AUC, see Materials and methods) is computed for HLA-specific diffRBM units' scores of immunogenic and non-immunogenic held-out peptides. (**B**) Performance of diffRBM units, full RBM, background RBM, and other methods, for the HLA-A*02:01 dataset. Semi-supervised methods (red) are trained only on immunogenic (or presented) peptides. Supervised methods (green) are trained with immunogenic and non-immunogenic peptides. 'DiffRBM units (difference)' is intermediate, exploiting the annotation of peptides as immunogenic and non-immunogenic *a-posteriori* (but it is not trained for the discrimination task). (**C–D**) Same as B, for HLA-B*07:02 (**C**) and HLA-B*35:01 (**D**). All AUC values are the averaged over 50 train/test set partitions, and error bars give the corresponding standard deviation (Materials and methods).

The online version of this article includes the following figure supplement(s) for figure 4:

**Figure supplement 1.** Comparison of performance of differential models of immunogenicity.

**Figure supplement 2.** Score comparison between immunogenic peptides and peptides from the human proteome.

**Figure supplement 3.** Leave-one-organism-out cross-validation for HLA-A*02:01-specific model (Materials and methods).

**Figure supplement 4.** Further comparison of diffRBM and RBM scores.

**Figure supplement 5.** Hyperparametric search for the classifier of immunogenicity.

**Figure supplement 6.** Performance of differential models of immunogenicity with sample reweighting.

distinguish immunogenic from non-immunogenic peptides. We therefore assigned scores of immunogenicity based on the diffRBM units to held-out test sets of peptides (*Figure 4A*) and we measured the score's ability to discriminate HLA-specific immunogenic peptides from non-immunogenic ones sharing the same HLA specificity in terms of the area under the receiver operating characteristic curve (AUC), see inset of *Figure 4A* and Materials and methods.

In *Figure 4B*, we compare the diffRBM units' AUC for the HLA-A*02:01-specific model to the AUC of the full RBM, that includes the background model in its predictions and hence yields a joint representation of the enriched patterns and of the background constraints. The AUC of discrimination progressively decreases, as expected, from the model part that disentangles enriched patterns from the background constraints (diffRBM units), to the full RBM capturing both, to the model fit to the background constraints only (background RBM). (In fact, there is no reason why background RBM should predict anything at all unless presentability and immunogenicity are correlated). This trend indicates that learning the background constraints that are shared by immunogenic and non-immunogenic peptides along with the sequence pattern enrichment distinctive of immunogenicity can act as a confounding factor when we look for the features that characterize and distinguish immunogenic peptides. Also simpler differential models relying on the independent-site assumption, while returning lower AUCs than RBM-based models, exhibit a decrease in AUC between the differential part and the full model (*Figure 4—figure supplement 1*). The AUC values and trend remain stable when we compare scores assigned to immunogenic peptides and to peptides from the human proteome (*Figure 4—figure supplement 2A*) and when we score immunogenic and non-immunogenic peptides from the same organism having trained the models on the immunogenic peptides from all the other organisms (*Figure 4—figure supplement 3*).

Conversely, the diffRBM units are not designed to capture the background constraints (here associated to presentability). As a result, they cannot successfully discriminate presentable antigens from generic peptides that are predominantly non-presentable (like peptides randomly drawn from the human proteome), in contrast to background RBM and the full RBM (*Figure 4—figure supplement 2B*). The same trends of discrimination performance are consistently found across the 3 HLA alleles considered (*Figure 4C–D*, *Figure 4—figure supplement 1*, *Figure 4—figure supplement 2*).

The prediction by the background model (trained on peptides probed for their binding to a given HLA-I) provides no clear signal that already the binding affinity to the HLA can discriminate immunogenic peptides (average AUC = 0.55 for HLA-A*02:01, 0.50 for HLA-B*07:02, 0.53 for HLA-B*35:01, see *Figure 4*). To further check this prediction, we scored HLA-A*02:01 peptides by their binding affinity to the HLA through NetMHCpan4.1 (*Reynisson et al., 2020*) and found a comparable difference in score distributions between immunogenic and non-immunogenic peptides (AUC = 0.54 for HLA-A*02:01, 0.48 for HLA-B*07:02, 0.53 for HLA-B*35:01). Our observation is in line with a recent large-scale mapping of killer T-cell recognition of candidate neopeptides at high HLA affinity (*Kristensen et al., 2022*), which did not find a significantly different distribution of HLA-binding NetMHCpan scores between immunogenic *vs* non-immunogenic neopeptides. Other studies, however, have suggested that immunogenic peptides bind more strongly to the HLA compared to non-immunogenic ones, both in the case of viral epitopes (*Croft et al., 2019*; *Buckley et al., 2022*) and neo-epitopes (*Bjerregaard et al., 2017*; *Buckley et al., 2022*). More work is needed in the future to clarify the link between binding affinity to the HLA and immunogenicity.

## A deeper classifier reaches optimal performance, but diffRBM stays comparable

To perform a task of classification of immunogenic *vs* non-immunogenic peptides, it is more effective to leverage the information in both sets of immunogenic and non-immunogenic peptides. In our framework of differential learning, this can be done by training a second set of differential units on the non-immunogenic peptides, which learns differences in their amino acid statistics with respect to the background. The difference of the scores of the diffRBM units trained on immunogenic peptides and the ones trained on non-immunogenic peptides plays as well the role of a score of immunogenicity, expected to be positive when evaluated on immunogenic peptides and negative on non-immunogenic peptides (Materials and methods). We use it to classify immunogenic *vs* non-immunogenic peptides, resorting to the AUC to measure its classification performance (*Figure 4B–D*).

Exploiting the availability of both immunogenic and non-immunogenic peptides, we also trained a deep neural-network-based classifier. The classifier architecture was optimized among several ones of different depth and width (Materials and methods, *Figure 4—figure supplement 5*) and reaches a performance quantified by a cross-HLA average AUC = $0.71 \pm 0.02$ (see 'optimal classifier' in *Figure 4B–D*; here and in the following uncertainties are estimated over several training-test partitions, see Materials and methods), higher than the best linear classifier (AUC = $0.66 \pm 0.02$). The AUC of the optimal classifier sets the maximal predictive performance that can be achieved, with the datasets under consideration, by a supervised method that is trained to discriminate immunogenic and non-immunogenic antigens. Its value (AUC = $0.71 \pm 0.02$) indicates that the predictability of immunogenicity from peptide sequences is limited, both by data availability and by the fact that sequence patterns along the peptide are not the only determinant of a positive T-cell response. In the future, more exhaustive assessments of peptide immunogenicity should account for the composition of cognate TCRs, peptide expression levels as well as the regulatory dynamics underlying T-cell response in physiological conditions. The performance of the diffRBM units' scores (AUC = $0.69 \pm 0.02$) is slightly lower than the one of the optimal classifier but higher than the one of the linear classifier (*Figure 4B–D*). We emphasize that the deep classifier is trained in a supervised way (*i.e.* using information on the two 'labels', immunogenic and non-immunogenic) while our RBM-based approach in *Figure 4* is 'semi-supervised' because its training requires knowledge of a portion of the 'labels' (the antigens labelled as 'immunogenic') only.

Finally, we checked that the performance of diffRBM compares favorably to established sequence-based methods for immunogenicity prediction (*Calis et al., 2013*; *Schmidt et al., 2021*; *Riley et al., 2019*), for more details see Supporting Information - Appendix 1 - Comparison of performance with existing tools.

## DiffRBM model for T-cell-specific binding

To train diffRBM models of epitope specificity, we first collected from the VDJdb database (*Shugay et al., 2018*; *Bagaev et al., 2020*) datasets of TCRs specific to 4 epitopes (Materials and methods): the $M1_{58}$ peptide from the influenza virus (with sequence GILGFVFTL), the $pp65_{495}$ from the human cytomegalovirus (CMV, with sequence NLVPMVATV), the $BMLF1_{280}$ peptide from the Epstein-Barr virus (EBV, with sequence GLCTLVAML), the peptide from the Spike protein $S_{269}$ from Sars-Cov-2 (with sequence YLQPRTFLL). We limited the search to sequences of the $\beta$ chain of the TCR (TCRβ), where the sites of binding to the antigen are concentrated in a region called complementarity determining region 3 (CDR3β).

The background dataset, in this case, is meant as a typical bulk TCRβ repertoire in normal conditions (*Figure 5—figure supplement 1*). In particular we take, to train background RBM, the repertoire of a hypothetical universal donor that was constructed by *Isacchini et al., 2021* from the TCRβ repertoires of the large scale study *Emerson et al., 2017* (Materials and methods). The sequence features captured by background RBM concern germline-encoded amino acid usage related to stability and binding constraints (as is the case for the two conserved residues cysteine and phenylalanine delimiting the CDR3β region), as well as additional biases in amino acid usage stemming from VDJ recombination and thymic selection.

After having trained the background RBM, we train a set of diffRBM units on each set of epitope-specific CDR3β (*Figure 5A*, *Figure 5—figure supplement 1*). By design, these diffRBM units capture antigen-driven convergent sequence features that have been documented in connection to epitope specificity (*Dash et al., 2017*; *Glanville et al., 2017*; *Meysman et al., 2019*; *Pogorelyy and Shugay, 2019b*; *Thakkar and Bailey-Kellogg, 2019*; *Mayer-Blackwell et al., 2021*; *Valkiers et al., 2021*). As such, similarly to the model of antigen immunogenicity, the diffRBM units can predict contact sites along the CDR3β (*Figure 1E*) and classify specific receptor against generic, predominantly non-specific ones (*Figure 1F*).

## DiffRBM predicts CDR3-β contact positions with the peptide

It has been discussed that convergent features in receptors responding to same antigens have structural interpretation in terms of interactions across the peptide-TCR interface (*Dash et al., 2017*; *Glanville et al., 2017*) and that the TCR contact regions are dominated by CDR3β residues (*Glanville et al., 2017*; *Ostmeyer et al., 2019*; *Milighetti et al., 2021*; see also *Figure 1E*). Starting from the

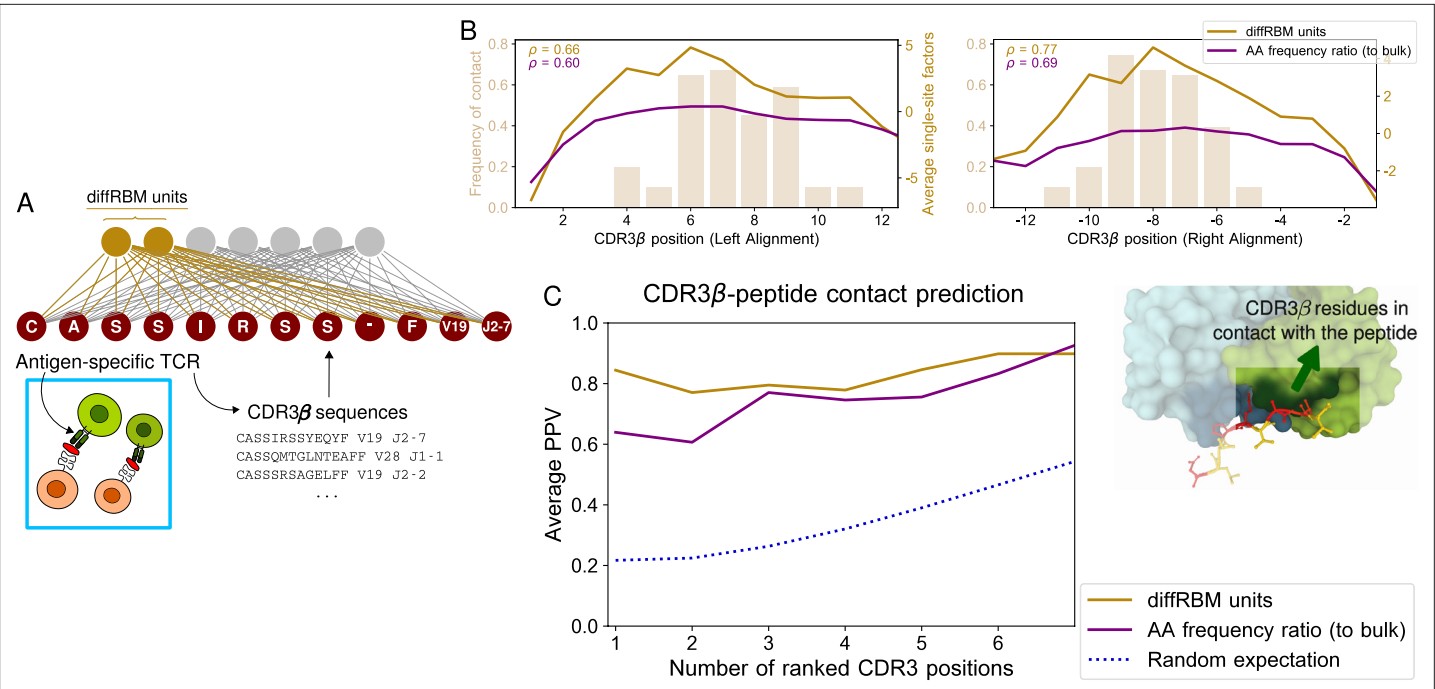

**Figure 5.** DiffRBM model of TCR epitope specificity and structural interpretation. (**A**) DiffRBM units are learnt from CDR3β sequences of antigen-specific TCRs. (**B**) Contact frequency distribution (bars) with peptide at each CDR3β position, across 12 structures (2 for YLQPRTFLL, 3 for NLVPMVATV, 1 for GLCTLVAML, 6 for GILGFVFTL). CDR3β positions are given as distances to either the left or right anchor sites. Contacts are sites with distance ≤4 Å between CDR3β and peptide. Magnitude of single-site factors based on the diffRBM units or the amino acid frequency ratio (of peptide-specific sequences relative to bulk-repertoire sequences) averaged over the 12 CDR3β are shown as lines. (**C**) PPV of CDR3β-peptide contact positions, averaged over the 12 structures, using single-site factors from the peptide-specific models (diffRBM or amino-acid frequency ratios). PPVs are reweighed by CDR3β sequence similarity (Materials and methods, *Figure 5—figure supplement 3A–B*).

The online version of this article includes the following figure supplement(s) for figure 5:

**Figure supplement 1.** Schematic summary of the construction of a diffRBM model of TCR epitope-specificity.

**Figure supplement 2.** Hyperparametric search for the diffRBM model of TCR specificity.

**Figure supplement 3.** Prediction of CDR3β contact positions with the peptide.

available TCR-pMHC structures already analyzed, we looked at the peptide-TCR contacts lying in the CDR3β region of the TCR, focusing on the structures involving the four epitopes under consideration (12 structures in total, see Materials and methods). *Figure 5B* shows the distribution of contacts along the CDR3β positions, which consists, as already observed in *Glanville et al., 2017*; *Ostmeyer et al., 2019*; *Milighetti et al., 2021*, of stretches of 3–5 contiguous amino acids in the central part of the CDR3 (6–8 positions from the left and right anchor).

We estimated the diffRBM single-site factors for the CDR3β sequences from the available structures. Their average value across CDR3β sequences concentrates on the CDR3β central positions and well correlates with the contact frequency distribution (*Figure 5B*). We next ranked the positions by the single-site factor magnitude and we took top ranking positions as positions of predicted contact. *Figure 5C* shows the PPV averaged over the 12 available structures (Materials and methods). Similarly to the predictions for peptides (*Figure 2D*), the predictive power of the diffRBM units is superior to the position-specific amino acid frequency ratio between the antigen-specific and the bulk-repertoire set of receptors, and is substantially higher than the random expectation (the p-value of this difference at the first ranked position is $1.6 \times 10^{-5}$, see Materials and methods). The PPV trend stays robust varying the distance cutoff from 4 up to 5 Å (the value chosen to determine peptide-TCR contacts in other work *Calis et al., 2012*; *Glanville et al., 2017*; *Ostmeyer et al., 2019*; *Milighetti et al., 2021*), see *Figure 5—figure supplement 3D*.

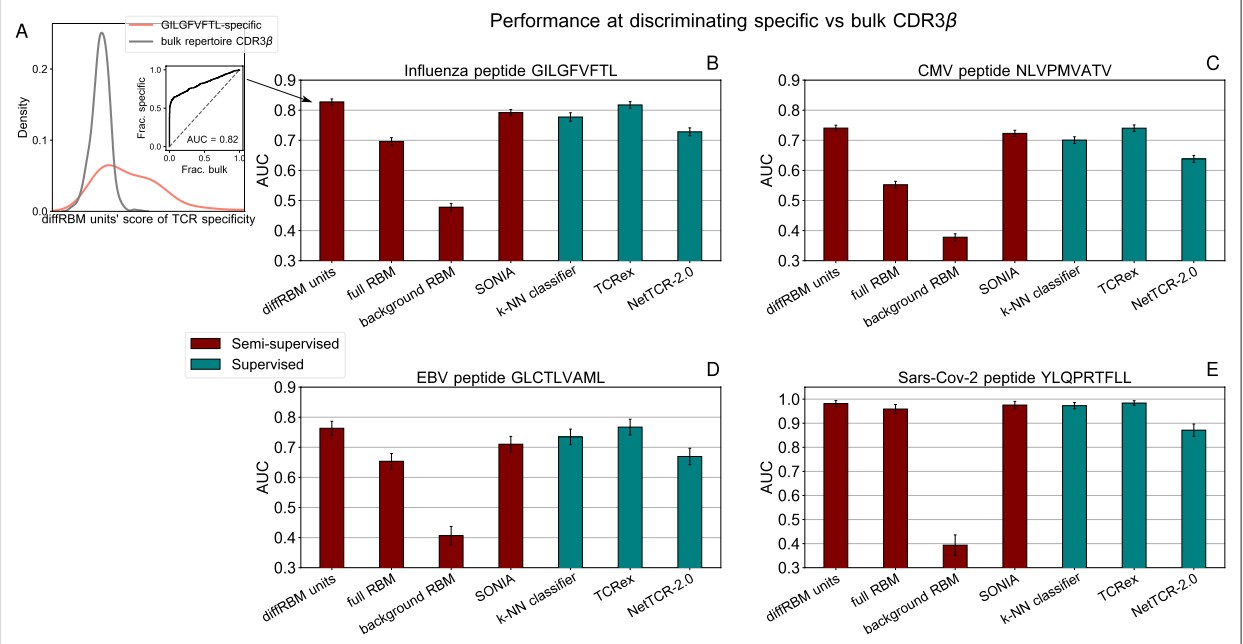

**Figure 6.** Performance at discriminating antigen-specific from generic T-cell receptors. (**A**) For a given epitope model (*e.g.* the Influenza epitope GILGFVFTL), we assign diffRBM units' scores to held-out sets of antigen-specific CDR3β and generic CDR3β from the bulk repertoire, and we measure the discrimination performance via the Area Under the Curve (AUC), see Materials and methods. (**B**) AUC of the diffRBM units, full RBM, background RBM and other methods trained and tested on CDR3β sequences specific to the Influenza epitope GILGFVFTL. (**C–E**) The performance assessment illustrated in **A–B** is repeated for the models of specific response to the CMV epitope NLVPMVATV (**C**) EBV epitope GLCTLVAML (**D**), and the Sars-Cov-2 epitope YLQPRTFLL (**E**). AUC values shown are the average over 50 partitions into training and test sets and error bars give the corresponding standard deviation (Materials and methods).

The online version of this article includes the following figure supplement(s) for figure 6:

**Figure supplement 1.** Comparison of performance of differential models of TCR specificity.

**Figure supplement 2.** Comparison of performance of differential models of TCR specificity with different background datasets.

**Figure supplement 3.** Hyperparametric search of the optimal $k$ for the $k$-NN algorithm.

**Figure supplement 4.** Comparison of performance of models of TCR specificity without V and J type.

## DiffRBM discriminates specific receptors

We tested the power to discriminate receptors specific to the 4 epitopes under consideration (GILGFVFTL, NLVPMVATV, GLCTLVAML, YLQPRTFLL) from generic sequences drawn from a bulk repertoire (background dataset). These can be seen as a proxy for non-specific receptors, since the large majority of them is not expected to respond to a specific epitope. We measured the performance at discriminating specific from generic receptors by the AUC (*Figure 6A*) and the results for the 4 epitope-specific models are reported in *Figure 6B–E*, *Figure 6—figure supplement 1*. We observed the same trend described for the models of immunogenicity, whereby the AUC of discrimination for the diffRBM units is consistently higher than the one for the full RBM. Also in this context, singling out the sequence features associated to epitope specificity, as the differential units do, enhances the model's predictive performance compared to the full RBM, where the information on those features is added to the background constraints. Any discrimination power is lost when using the background RBM, as it should, since it has no information on epitope specificity.

## DiffRBM reaches state-of-the-art performance

For the sake of comparison of diffRBM to other tools, we considered another generative model of antigen-specific repertoires, SONIA (*Elhanati et al., 2014*; *Sethna et al., 2020*), and a series of methods trained to discriminate target-specific from unspecific receptors (labeled as 'supervised' in *Figure 6B–E*), which include: a baseline method to determine the predictive power achievable from TCR sequence similarity alone (a $k$-Nearest Neighbours classifier, $k$-NN, see Supporting Information

- Appendix 3 - Alternative approaches tested and *Figure 6—figure supplement 3*); the state-of-the-art supervised predictors of TCR specificity TCRex (*Gielis et al., 2019*) and NetTCR-2.0 (*Montemurro et al., 2021*). As shown in *Figure 6B–E*, the performance of the diffRBM units is fully comparable to the most performant among the state-of-the-art supervised methods, TCRex (both with an average AUC across peptides of $0.83 \pm 0.01$), and higher than SONIA, $k$-NN (AUC = $0.80 \pm 0.01$ for both), and NetTCR-2.0 (AUC = $0.73 \pm 0.01$). Similarly, when computing the average precision (AP), diffRBM (with AP averaged over the 4 peptides of $0.87 \pm 0.01$) works as well or slightly better than TCRex (AP = $0.86 \pm 0.01$) and better than NetTCR2.0 (AP = $0.76 \pm 0.01$), see Supporting Information - Appendix 1 - Comparison of performance with existing tools.

## Discussion

We have introduced a machine-learning framework, diffRBM, based on the probabilistic graphical model of Restricted Boltzmann Machines to address two important and complementary issues in the context of the immune response, namely the modeling of antigen immunogenicity and epitope-specific TCRs. Our approach relies on a 'background' dataset of large size, on which a background RBM model is trained, and a small subset of selected data on which some additional hidden units of the RBM, the diffRBM units, are learnt (*Figure 1*, *Figure 1—figure supplement 1*).

First, we applied diffRBM to model the ability of an antigen to trigger a positive T-cell response. We showed that diffRBM units encode several features relevant to antigen immunogenicity, leading to biologically interpretable predictions. For instance, diffRBM allows us to estimate which peptide positions are likely to be in contact with the TCR (*Figure 2*), outperforming immunogenicity predictors tools that treat peptide sites as independent of each other (*Calis et al., 2013*). Our model construction does not require an ad hoc selection of residue positions (like in *Schmidt et al., 2021*) or properties (like in *Chowell et al., 2015*) that are assumed to be predictive of immunogenicity. Rather, the diffRBM units directly learn patterns of enrichment in certain biophysical properties, such as aromaticity, across peptide positions. As such, diffRBM probabilistic scores can be used to predict mutational costs in terms of TCR reactivity (*Figure 3*) and to distinguish immunogenic from non-immunogenic peptides, with performances comparable to supervised classifiers (*Figure 4*).

Second, we have trained differential models from datasets of epitope-specific T-cell receptors, and have successfully tested the models' power to identify the CDR3β residues that bind to the antigen (*Figure 5*). DiffRBM provides insight into the structural basis of this process, and helps discriminate epitope-specific from generic receptors (*Figure 6*). Our model performs as well as the state-of-the-art methods for distinguishing few antigen-specific receptors out of the bulk repertoire, both in terms of AUC and of average precision (*Figure 6*, Section Appendix 1 - 'Comparison of performance with existing tools'). Performing well on this task is important as the fraction of the TCR repertoire reactive to a given epitope is expected to range between $10^{-6}$ and $10^{-4}$ (*Yates, 2014*).

### Main differences of diffRBM with existing methods

In the following we discuss the main differences of diffRBM with state-of-the-art predictors for peptide immunogenicity and the ability of a TCR to bind to a given antigen. Most methods for TCR specificity prediction (*Gielis et al., 2019*; *Montemurro et al., 2021*; *Weber et al., 2021*) are classifiers built from two sets of labeled data: the positive samples containing the receptors with the given specificity and the negative ones containing receptors specific to other antigens. The diffRBM framework is semi-supervised, as negative samples, whose construction is somewhat arbitrary, are not needed.

In the context of antigen immunogenicity modeling, the diffRBM approach is specific to an HLA type, contrary to existing pan-HLA predictors (*Schmidt et al., 2021*). This choice has the advantage of better capturing the HLA-specific amino-acid usage associated with presentation in the background model, and of being robust against cross-HLA imbalances in the available data (HLA-A*02:01 is much better represented than other alleles). Similarly, our diffRBM models predict TCRs binding to a given peptide, while other methods (*Springer et al., 2020*; *Weber et al., 2021*; *Milighetti et al., 2021*) are predictive across potentially any peptide.

## DiffRBM is parsimonious

Even for the HLA alleles and epitopes we considered, available datasets cover only a small fraction of HLA-specific peptides that have the potential to be immunogenic and TCR that can recognize given peptides, and pool together data obtained through a heterogeneous set of T-cell response assays, a source of additional noise in the data (Materials and methods). Such limitations pinpoint the need for approaches that are parsimonious in terms of training data. For instance, our search for an optimal neural network classifier of immunogenicity returned an architecture with a rather modest depth (only one hidden layer, *Figure 4—figure supplement 5*). DiffRBM is tailored for such data-limited situations, and, yet, is able to capture interactions between residues that cannot be modeled by simpler approaches (*Figure 2*, *Figure 5*, *Figure 2—figure supplement 3*, *Figure 4—figure supplement 1*, *Figure 5—figure supplement 3*, *Figure 6—figure supplement 1*).

## DiffRBM is a generative model

It can be used to produce new, putative sequences of antigens or TCRs with desired properties, testable in experimental setups, with potential applications in vaccine design and TCR engineering for therapeutic purposes. The probabilistic distribution encoded by the model allows for estimating the diversity of the data space (through its entropy, Appendix 4), and the loss of diversity due to selection. In the case of epitope-specific repertoires (*Appendix 4—figure 1*), this reduction appears rather modest, consistently with previous analyses on epitope-specific CDR3β data (*Sethna et al., 2020*), with the exception of the YLQPRTFLL-specific repertoire.

## DiffRBM is purely sequence-based

The fast growth of peptidomic and immune repertoire sequence data, many of them produced in clinical settings for personalized medicine purposes, calls for the development of sequence-based modeling approaches. These are computationally faster and more broadly applicable than structure-informed methods. Structural features, such as the conformational arrangement of the peptide within the HLA binding groove and of TCR chains, have been shown to be important to predict a positive T-cell response, both at the level of peptide binding to the MHC (*Riley et al., 2019*) and TCR binding to the pMHC complex (*Lin et al., 2021*). However, these methods are often limited to TCR-peptide pairs of known crystal structures, narrowing down the scope of applications. To circumvent this limitation, a peptide-threading procedure into pre-defined template structures was proposed (*Riley et al., 2019*). This strategy is made possible by the homogeneity of the backbone conformation across nonamer peptides presented by the same HLA-I, but might become problematic with peptides of different lengths or for HLA types admitting different binding modes with the peptide (*Gfeller et al., 2018*).

There are many transfer-learning approaches in machine learning aimed at obtaining domain-invariant representations that enable portability across domains. From this point of view, several works have considered the RBM architecture, see for instance (*Zhang, 2011*; *Wei and Pal, 2011*). Our objective here is somewhat different: learning the statistical differences between background and selected data allows us not only to reach very good performance when data is scarce, but also to capture enriched molecular motifs in the selected data. The model parameters learnt in this way allow us to extract biologically interpretable predictions, such as contact sites and mutation costs in terms TCR reactivity.

## Biomedical applicability of diffRBM

Our computational predictions of antigen immunogenicity can be used to propose candidate antigens for experimental validation and vaccine design. Similarly, they can be used to identify potentially immunogenic neoantigens to target in immune-based cancer therapies, or to detect phenomena of immunoediting in cancer (*Łuksza et al., 2022*). To adapt diffRBM to neoantigen modeling, it may need to be re-trained on neoantigen-specific training sets (rather than on IEDB as done here), given the substantial variation in method performance and in the immunogenicity-related predicted features between pathogenic and cancer data (*Buckley et al., 2022*). As a preliminary assessment of diffRBM performance in a cancer setting, we assigned scores to the HLA-A*02:01-presented neoantigens from the TESLA dataset (*Wells et al., 2020*; 11 immunogenic and 227 non-immunogenic peptides). We obtained an AUC of immunogenic *vs* non-immunogenic discrimination, differing significantly from the

random expectation (0.6 for the diffRBM units' scores and 0.69 for the difference of diffRBM units' scores).

## Future extensions of diffRBM

This diffRBM model of specific TCR response can be formulated in an equivalent way for the CDR3 region on the other chain of TCRs ($\alpha$ chain). In this case, one uses CDR3$\alpha$ sequences from healthy TCR repertoires and samples of antigen-specific CDR3$\alpha$ as respectively the background and the selected dataset. DiffRBM models for the $\alpha$ chain reach a discrimination performance comparable to the one for the $\beta$ chain (**Meysman et al., 2022**). Similarly, with single-cell TCR sequencing data becoming increasingly available, our approach could also be extended to model pairs of TCR $\alpha$ and $\beta$ chains, which have been suggested to play a synergistic role in determining antigen specificity (**Carter et al., 2019**; **Montemurro et al., 2021**; **Milighetti et al., 2021**). An important direction for future work could be to improve the modeling strategy presented by leveraging structural information on the TCR-pMHC complex and its estimated binding energy along with sequences, as investigated by **Riley et al., 2019**; **Lin et al., 2021**; **Milighetti et al., 2021**; **Karnaukhov et al., 2022**, especially given the expected increase in the number of available crystallographic structures.

Last of all, diffRBM provides a potentially general and flexible approach, which can be used in analyzing directed evolution protocols (**Jäckel et al., 2008**; **Packer and Liu, 2015**; **Arnold, 2018**) and SELEX experiments (**Ellington and Szostak, 1990**; **Tuerk and Gold, 1990**; **Sola et al., 2020**), where each round performs a selection of a subset of molecules from the previous round (the 'background'). These potential applications require novel efforts of data pre-processing and model training. For instance, the present setting, in which the two learning steps are done sequentially, needs to be modified when the baseline and the specific features are entangled.

# Materials and methods
## DiffRBM architecture

The core idea of what we refer to as a 'differential' probabilistic model is to learn a distribution for sequence data $\boldsymbol{\sigma}$ with the parametric form:

$$P(\boldsymbol{\sigma}) = \frac{1}{Z}e^{-\mathcal{H}(\boldsymbol{\sigma})} \qquad \mathcal{H}(\boldsymbol{\sigma}) = \mathcal{H}^{\mathrm{b}}(\boldsymbol{\sigma}) + \mathcal{H}^{\mathrm{d}}(\boldsymbol{\sigma}) \tag{1}$$

where $Z$ is simply a normalization factor, $Z = \sum_{\sigma} e^{-\mathcal{H}(\sigma)}$. $\mathcal{H}^{\mathrm{b}}$ specifies the background distribution $P^{\mathrm{b}}$, learnt from the background dataset $\mathcal{D}^{\mathrm{b}}$, through $P^{\mathrm{b}}(\boldsymbol{\sigma}) = 1/Z^{\mathrm{b}}e^{-\mathcal{H}^{\mathrm{b}}(\sigma)}$, where $Z^{\mathrm{b}} = \sum_{\sigma} e^{-\mathcal{H}^{\mathrm{b}}(\sigma)}$. $\mathcal{H}^{\mathrm{d}}$ contains the parameters learnt on top of the background distribution from the selected data $\mathcal{D}^{\mathrm{s}}$. Here, we assume that $\mathcal{H}^{\mathrm{b}}$ and $\mathcal{H}^{\mathrm{d}}$ are parametrized in terms of a Restricted Boltzmann Machine (RBM; **Hinton, 2002**; **Hinton and Salakhutdinov, 2006**). Hence $P^{\mathrm{b}}(\boldsymbol{\sigma})$ can be written as:

$$P^{\mathrm{b}}(\boldsymbol{\sigma}) = \int \prod_{\mu=1}^{M^{\mathrm{b}}} dh_{\mu}^{\mathrm{b}} P^{\mathrm{b}}(\boldsymbol{\sigma}, h^{\mathrm{b}}) \quad P^{\mathrm{b}}(\boldsymbol{\sigma}, h^{\mathrm{b}}) \sim \exp\left(\sum_{i=1}^{N} g_i^{\mathrm{b}}(\sigma_i) - \sum_{\mu=1}^{M^{\mathrm{b}}} \mathcal{U}_{\mu}^{\mathrm{b}}(h_{\mu}^{\mathrm{b}}) + \sum_{i,\mu} h_{\mu}^{\mathrm{b}} w_{i\mu}^{\mathrm{b}}(\sigma_i)\right) \tag{2}$$

that is, in an RBM, the probability of the data $P^{\mathrm{b}}(\boldsymbol{\sigma})$ is expressed as the marginal of a joint probability over the data $\boldsymbol{\sigma}$ (the 'observed' sequences of length $N$) and a set of $M^{\mathrm{b}}$ 'hidden' units $\boldsymbol{h}^{\mathrm{b}}$, playing the role of coordinates of low-dimensional representations of the data. Thus, for an RBM as background:

$$\mathcal{H}^{\mathrm{b}}(\boldsymbol{\sigma}) = -\left(\sum_{i=1}^{N} g_i^{\mathrm{b}}(\sigma_i) + \sum_{\mu=1}^{M^{\mathrm{b}}} \Gamma_{\mu}^{\mathrm{b}}\left(I_{\mu}^{\mathrm{b}}(\boldsymbol{\sigma})\right)\right) \tag{3}$$

where we have set $I_{\mu}^{\mathrm{b}}(\boldsymbol{\sigma}) = \sum_i w_{i\mu}^{\mathrm{b}}(\sigma_i)$ and $\Gamma_{\mu}^{\mathrm{b}}\left(I_{\mu}^{\mathrm{b}}(\boldsymbol{\sigma})\right) = \log \int dh_{\mu}^{\mathrm{b}} e^{-\mathcal{U}_{\mu}^{\mathrm{b}}(h_{\mu}^{\mathrm{b}}) + h_{\mu}^{\mathrm{b}} I_{\mu}^{\mathrm{b}}(\boldsymbol{\sigma})}$. *Equations 2 and 3* contain the following parameters: a set of single-site fields $g_i^{\mathrm{b}}(\sigma_i)$, capturing the amino acid usage at each sequence position, a potential $\mathcal{U}_{\mu}^{\mathrm{b}}$ for each hidden unit $h_{\mu}^{\mathrm{b}}$ and a set of parameters $w_{i\mu}^{\mathrm{b}}$, called weights, connecting the sites of observed sequences to each hidden unit. Their values are learnt from the background dataset $\mathcal{D}^{\mathrm{b}}$ by maximizing the log-likelihood:

$$\frac{1}{|\mathcal{D}^{\mathrm{b}}|} \sum_{\boldsymbol{\sigma} \in \mathcal{D}^{\mathrm{b}}} \log P^{\mathrm{b}}(\boldsymbol{\sigma}) = \left\langle \log P^{\mathrm{b}}(\boldsymbol{\sigma}) \right\rangle_{\mathcal{D}^{\mathrm{b}}} \tag{4}$$

After the background distribution has been learnt, we learn the differential units of the diffRBM architecture, specified by the probability distribution:

$$P(\boldsymbol{\sigma}, h^{\mathrm{b}}, h^{\mathrm{d}}) \sim \exp\left( \sum_{i=1}^{N} \left( g_i^{\mathrm{d}}(\sigma_i) + g_i^{\mathrm{b}}(\sigma_i) \right) - \sum_{\mu'=1}^{M^{\mathrm{d}}} \mathcal{U}_{\mu'}^{\mathrm{d}}(h_{\mu'}^{\mathrm{d}}) - \sum_{\mu=1}^{M^{\mathrm{b}}} \mathcal{U}_{\mu}^{\mathrm{b}}(h_{\mu}^{\mathrm{b}}) + \sum_{i,\mu'} h_{\mu'}^{\mathrm{d}} w_{i\mu'}^{\mathrm{d}}(\sigma_i) + \sum_{i,\mu} h_{\mu}^{\mathrm{b}} w_{i\mu}^{\mathrm{b}}(\sigma \tag{5}$$

see also *Figure 1—figure supplement 1*. Hence:

$$\mathcal{H}^{\mathrm{d}}(\boldsymbol{\sigma}) = -\left( \sum_{i=1}^{N} g_i^{\mathrm{d}}(\sigma_i) + \sum_{\mu'=1}^{M^{\mathrm{d}}} \Gamma_{\mu'}^{\mathrm{d}}\left( I_{\mu'}^{\mathrm{d}}(\boldsymbol{\sigma}) \right) \right) \tag{6}$$

where $I_{\mu'}^{\mathrm{d}}(\boldsymbol{\sigma}) = \sum_i w_{i\mu'}^{\mathrm{d}}(\sigma_i)$ and $\Gamma_{\mu'}^{\mathrm{d}}\left( I_{\mu'}^{\mathrm{d}}(\boldsymbol{\sigma}) \right) = \log \int dh_{\mu'}^{\mathrm{d}} \, e^{-\mathcal{U}_{\mu'}^{\mathrm{d}}(h_{\mu'}^{\mathrm{d}}) + h_{\mu'}^{\mathrm{d}} I_{\mu'}^{\mathrm{d}}(\boldsymbol{\sigma})}$. The parameters featuring in $\mathcal{H}^{\mathrm{d}}(\boldsymbol{\sigma})$, defining the diffRBM units, are learnt from the dataset $\mathcal{D}^{\mathrm{s}}$ by maximizing:

$$\frac{1}{|\mathcal{D}^{\mathrm{s}}|} \sum_{\boldsymbol{\sigma} \in \mathcal{D}^{\mathrm{s}}} \log P(\boldsymbol{\sigma}) = \left\langle \log P(\boldsymbol{\sigma}) \right\rangle_{\mathcal{D}^{\mathrm{s}}} \quad \text{with} \quad P(\boldsymbol{\sigma}) = \int \prod_{\mu=1}^{M^{\mathrm{b}}} dh_{\mu}^{\mathrm{b}} \prod_{\mu'=1}^{M^{\mathrm{d}}} dh_{\mu'}^{\mathrm{d}} P(\boldsymbol{\sigma}, h^{\mathrm{b}}, h^{\mathrm{d}}) \tag{7}$$

In summary, the diffRBM architecture is equivalent to an RBM with $M^{\mathrm{b}} + M^{\mathrm{d}}$ hidden units with overall observed fields $g_i^{\mathrm{b}} + g_i^{\mathrm{d}} : g_i^{\mathrm{b}}$, the weights $w_{i\mu}^{\mathrm{b}}$ and potentials $\mathcal{U}_{\mu}^{\mathrm{b}}$ for the first $M^{\mathrm{b}}$ units are learnt from the background data, then they are kept fixed, and $g_i^{\mathrm{d}}$, the weights $w_{i\mu'}^{\mathrm{d}}$ and potentials $\mathcal{U}_{\mu'}^{\mathrm{d}}$ for the additional $M^{\mathrm{d}}$ units are learnt from the selected data (*Figure 1—figure supplement 1*, Appendix 2).

The predictions of peptide immunogenicity or epitope specificity rely on the assignment to sequences $\boldsymbol{\sigma}$ of scores. Using *Equation 1*, the score of the full RBM is given by the log-likelihood:

$$\mathcal{L}(\boldsymbol{\sigma}) = \log P(\boldsymbol{\sigma}) = -\mathcal{H}^{\mathrm{b}}(\boldsymbol{\sigma}) - \mathcal{H}^{\mathrm{d}}(\boldsymbol{\sigma}) + constant \tag{8}$$

where the *constant* stands for a sequence-independent term coming from the partition function. Analogously, the background RBM score and the diffRBM units' score are respectively:

$$\mathcal{L}^{\mathrm{b}}(\boldsymbol{\sigma}) = -\mathcal{H}^{\mathrm{b}}(\boldsymbol{\sigma}) + constant \tag{9}$$

$$\mathcal{L}^{\mathrm{d}}(\boldsymbol{\sigma}) = -\mathcal{H}^{\mathrm{d}}(\boldsymbol{\sigma}) + constant \tag{10}$$

Software-wise diffRBM is coded via additional functions to execute a differential learning on top of the RBM Python implementation from *van der Plas et al., 2023* and is available at https://github.com/cossio/diffRBM (copy archived at *Fernandez-de-Cossio-Diaz, 2023*). The codes used for its application to modeling antigen immunogenicity and TCR specificity are downloadable from https://github.com/bravib/diffRBM_immunogenicity_TCRspecificity (copy archived at *Bravi, 2023*).

## Data collection

### Sequence datasets for the immunogenicity model

Differential models of immunogenicity were trained on sets of immunogenic peptides collected from the Immune Epitope Database (IEDB) (*Vita et al., 2019*), where the database entries were filtered through the following steps. Firstly, the curated set of HLA ligands tested in T cell assays was downloaded from IEDB (file *tcell_full_v3.csv* from http://www.iedb.org/database_export_v3.php, accessed in December 2021). We selected from this file linear, human peptides with a given HLA restriction (*e.g.* HLA-A*02:01), limiting the search to peptides of length 8–11 amino acids like in *Bravi et al., 2021b* and presented by HLA of class I (*i.e.* targeted epitopes of killer T cells). Following *Calis et al., 2013*, we required the peptide (and not the full protein or the pathogen) to be the first immunogen (by setting the field *Antigen Epitope Relation* = 'Epitope') and we excluded T-cell response experiments with a restimulation step (by discarding 'Restimulation in vitro' from the field In Vitro Process Type). Immunogenic peptides were finally identified as the peptides for which positive responses by

T cells were reported while negative ones were absent (field *Qualitative Measure* marked as 'Positive' or 'Positive-High' and never as 'Negative'). Equivalently, non-immunogenic peptides were identified as the peptides for which negative responses by T cells were reported while positive ones were absent (field *Qualitative Measure* set to 'Negative' and never to 'Positive', 'Positive-High', 'Positive-Intermediate' or 'Positive-Low'). To avoid oversampling, we removed duplicate entries. To check whether we needed an additional redundancy filtering, similarly to *Calis et al., 2013*, we applied a reweighting scheme *Morcos et al., 2011* that reweighs each sequence by the inverse of the number of other sequences that have more than 80% of similarity, and we found that the models' performance (*Figure 4—figure supplement 6*) is largely unchanged compared to the one without reweighting (*Figure 4*), indicating that there is no substantial need for additional sampling bias mitigation strategies. We choose only the HLA-I alleles for which the filtering steps just described allowed us to recover at least 200 immunogenic peptides and for which at least one TCR-pMHC structure was available in the Protein Data Bank (resulting in the choice of HLA-A*02:01, HLA-B*07:02 and HLA-B*35:01). The size of the final datasets of immunogenic peptides is: $|\mathcal{D}^s|$ = 1682 for HLA-A*02:01, $|\mathcal{D}^s|$ = 258 for HLA-B*07:02, $|\mathcal{D}^s|$ = 215 for HLA-B*35:01. Sets of non-immunogenic peptides consist of 2301 sequences (HLA-A*02:01), 807 (HLA-B*07:02), 166 (HLA-B*35:01).

To train the antigen presentation model (background model), we relied on the sets of 8–11 amino acid long peptides extracted from IEDB by the RBM-MHC algorithm as described in *Bravi et al., 2021b*, choosing the option of peptides from HLA binding affinity assays rather than mass spectrometry, to avoid biases in the amino acid statistics that might be due to this technique. The resulting training dataset sizes are $|\mathcal{D}^b|$ = 4265 for HLA-A*02:01, $|\mathcal{D}^b|$ = 1006 for HLA-B*07:02, $|\mathcal{D}^b|$ = 1211 for HLA-B*35:01. For consistency with the type of datasets used in RBM-MHC, scores of presentation from the algorithm NetMHCpan4.1 (*Reynisson et al., 2020*) are obtained with the option *-BA* (predictions from the training on binding assay data).

## Sequence datasets for the T-cell specificity model

Each differential model of specific T-cell binding to a given peptide was trained on TCRs experimentally validated to be specific to the peptide collected from the VDJdb database (*Shugay et al., 2018*; *Bagaev et al., 2020*) (file *vdjdb.txt* downloaded from https://vdjdb.cdr3.net in July 2021). We selected all the human TCRβ chains fully annotated with their V and J segment and labeled to be specific to the given peptide (for example, for the Influenza M1$_{58}$-specific model we set *antigen.epitope* = 'GILG-FVFTL'). We constructed the training sets from the CDR3β sequence and the V/J annotation of these entries, removing replicates. Their size is: $|\mathcal{D}^s|$ = 3464 (for the Influenza M1$_{58}$ model), $|\mathcal{D}^s|$ = 4548 (for the CMV pp65$_{495}$ model), $|\mathcal{D}^s|$ = 993 (for the EBV BMLF1$_{280}$ model), and $|\mathcal{D}^s|$ = 315 (for the Sars-Cov-2 S$^{269}$ model).

For the background model, we considered the dataset assembled by *Isacchini et al., 2021* pooling together unique TCRβ clones from the 743 donors of the cohort in *Emerson et al., 2017*, with a total of ~9 × 10$^7$ sequences. For training background RBM, we used a smaller, randomly subsampled dataset of 10$^6$ sequences that could be more easily handled. We also considered the dataset collected by *Britanova et al., 2016* to train a second, independent background model. This dataset contains about 3×10$^7$ CDR3β sequences. We excluded from this dataset any sequence associated to non-functional V or J genes, and randomly subsampled the resulting dataset to obtain 10$^6$ sequences for the training of the background RBM and 10$^4$ sequences to test it (with no identical sequences between training and test set).

## Data pre-processing and formatting

RBM and PWM-based approaches require sequence inputs of fixed length, hence we performed an alignment. Background datasets are aligned to obtain same-length sequences, following the alignment procedures described in *Bravi et al., 2021b* for peptides and *Bravi et al., 2021a* for CDR3β amino acid sequences. The length of the alignment is set to 9 (9 being the typical length of HLA-I ligands) and to 20 in the case of CDR3β. These alignments serve as seeds to learn Hidden Markov Model profiles of length 9 and 20, in such a way that the selected datasets can be aligned against the profile built from the corresponding background dataset (see *Bravi et al., 2021b* for more details). In the models for TCR sequences, the input combines the aligned CDR3β amino acid sequence to the V segment type and the J segment type, all converted into numerical values varying within an interval

of appropriate length (length = 21 for the CDR3β positions, standing for the 20 amino acids + 1 gap; length = 48 for the V type, length = 13 for the J type).

## Crystallographic structures from PDB

We downloaded, from the Protein Data Bank https://www.rcsb.org/ (*Berman et al., 2000*) as of February 2022, the TCR-pMHC crystallographic structures with 9 amino acid-long peptides where the HLA complex is HLA-B*35:01, HLA-A*02:01, or HLA-B*07:02. We excluded the structures with modified/non-peptidic epitopes and with incomplete TCR chains. As a result, we obtained 5 structures for HLA-B*35:01, 56 for HLA-A*02:01, and 1 for HLA-B*07:02. Some of the 56 HLA-A*02:01 structures describe TCRs in contact with the peptides we considered for the differential models of specific TCR response (3 for the Sars-Cov-2 epitope YLQPRTFLL, 3 for the CMV epitope NLVPMVATV, 1 for the EBV epitope GLCTLVAML, and 8 for the Influenza epitope GILGFVFTL).

For each structure, we estimated the positions along the peptide in contact with the TCR, using a standard cutoff at 4 Å (*Rossjohn et al., 2015*; *Schmidt et al., 2021*; *Lu et al., 2021*) between heavy atoms. The availability of structures is highly skewed toward the limited set of epitopes that have been the focus of several studies, hence our final list of peptides exhibits redundancy, with same or similar peptides in complex with different TCRs. If more than 1 structure contain the same peptide and have same contact positions, we retain only one of such structures (resulting in 4 structures for HLA-B*35:01, 41 for HLA-A*02:01, and 1 for HLA-B*07:02). If the same ligand is annotated with different contact positions, we keep these as different entries but we re-weight their contribution to the average PPV and the frequency of contact positions (section 'Contact prediction'). We followed the same steps to estimate CDR3β contacts with the peptide and peptide contacts with the HLA complex and to filter out redundant entries, opting for a slightly more restrictive cutoff distance (3.5 Å) for peptide-HLA contacts. Since distance cutoffs can vary with the van der Waals' radii for single atoms (*Sheriff et al., 1987*), we also monitored the robustness of our results to changes in the choice of the cutoff (*Figure 2—figure supplement 3D*, *Figure 2—figure supplement 4D*, *Figure 5—figure supplement 3D*). The list of all structures and corresponding estimated contacts is provided in *Figure 2—source data 1*.

## DiffRBM training and model selection

The first step of the diffRBM training consists of training the background model on the background dataset. For the model of immunogenicity, we trained allele-specific presentation models with an RBM architecture by running the RBM-MHC algorithm (*Bravi et al., 2021b*) on IEDB-derived peptide data (see section 'Sequence datasets for the immunogenicity model') with default parameters (10 hidden units, $\lambda_2^1 = 0.001$). The RBM-MHC algorithm internally aligns peptide sequences of its training dataset to the reference length of 9 amino acids; we used the same alignment routine to align the immunogenic peptides against the seed given by the RBM-MHC training data. The second step is training the diffRBM units on the selected datasets $\mathcal{D}^s$ (see section 'Sequence datasets for the immunogenicity model'). We divided these datasets into a training set with 80% of the data (used for training and for model selection) and a test set with the remaining 20% (used for model validation, see section 'Classification performance'), repeating this split 50 times.

Having fixed the background model, we used the largest available dataset of immunogenic sequences (the one for HLA-A*02:01) to perform model selection by cross-validation, as follows. We further divided randomly each of the 50 training sets into a set actually used for training and a validation set (with respectively 80% and 20% of the training set). We used this training/validation partitions to select optimal hyperparameters for the differential part (the number of hidden units and regularization penalty $\lambda_2^1$), by training diffRBM models on the training set at varying hyperparameters and monitoring the average diffRBM units' score (*Equation 10*) on the validation sets (*Figure 2—figure supplement 2A–B*). We also performed additional checks on the diffRBM units' AUC of immunogenic *vs* non-immunogenic discrimination with different hyperparameters (*Figure 2—figure supplement 2C*) and in a control case (*Figure 2—figure supplement 2D*).

For the model of T-cell specificity, the background is given by an RBM trained on a random subsample of $10^6$ CDR3β sequences from *Emerson et al., 2017* (section 'Sequence datasets for the T-cell specificity model'), choosing the optimal RBM architecture (100 hidden units, $\lambda_2^1 = 0.001$) by cross-validation (*Figure 5—figure supplement 2A–B*). For cross-validation, we used as validation set

another subsample of about $10^5$ CDR3β sequences from *Emerson et al., 2017*, chosen so that no overlap with the training set is present. The grid search for optimal hyperparameters for the differential part was carried out at fixed background model, using the largest dataset of peptide-specific receptors (NLVPMVATV) and partitioning it 50 times at random into training and validation sets (with respectively 80% and 20% of the data, *Figure 5—figure supplement 2C–E*).

## Classification performance

Having found the optimal diffRBM architecture (section 'DiffRBM training and model selection') for the immunogenicity model, for each of the 3 HLA types considered we trained 50 HLA-specific models on the original training sets (consisting of the 80% of the full datasets available) and we assessed their average performance over the corresponding 50 choices of the test set. In particular, we tested the ability of the HLA-specific immunogenicity models to identify new immunogenic peptides by the Receiver Operating Characteristic curve (ROC). For each of the 50 repetitions, we assigned scores of immunogenicity predicted by a given HLA-specific model (given by the diffRBM units' score $\mathcal{L}^d$ of *Equation 10*) to the sequences of the test set of positives (immunogenic peptides with the HLA type under consideration) and of negatives (a test set of non-immunogenic peptides presented by the same HLA). Varying the threshold score value to discriminate positives from negatives, we obtained the ROC curve describing the fraction of immunogenic peptides predicted by the models' scores, against the fraction of predicted non-immunogenic ones. We took the Area Under the Curve (AUC) as a metric of the models' ability to discriminate immunogenic from non-immunogenic peptides. We performed the same validation for all the RBM-based approaches (section 'DiffRBM architecture', *Figure 4*), using their corresponding output scores (*Equation 8* for the full RBM, *Equation 9* for background RBM). The performance of the full RBM obtained through the score (*Equation 8*) is by far and large equivalent to the one of an RBM with the same hyperparameters entirely trained, in one step only, from the selected dataset (*Figure 4—figure supplement 4*), showing that there is a gain in performance with the differential learning strategy only when we focus on the differential units and their parameters. For the diffRBM linear approach (section Appendix 3 - 'Alternative approaches tested'), the scores (*Equation 8*) and (*Equation 10*) contain, for the differential part, only fields $g_i^d(\sigma_i)$; in the PWM-based approach, we used (*Equation 18*) (see section Appendix 3 - 'Alternative approaches tested'); results from these approaches are reported in *Figure 4—figure supplement 1*.

We also performed a leave-one-organism-out cross-validation, whereby we divided peptides by the organism of origin, we held out as test set only the immunogenic and non-immunogenic peptides from the same organism and trained the models on the peptides from all the other organisms (*Figure 4—figure supplement 3*). We considered in the test sets only the organisms for which at least 15 immunogenic and non-immunogenic peptides could be retrieved from IEDB. HLA-A*02:01 is the only allele for which we found sufficient data for this validation.

Also negatives were randomly divided into 50 training and test sets. Training sets of negatives were used to train diffRBM units for non-immunogenic peptides (assigning scores that we will denote as $\mathcal{L}^{d,neg}(\boldsymbol{\sigma})$ to distinguish it from the scores of immunogenicity $\mathcal{L}^d(\boldsymbol{\sigma})$) and the classifiers of immunogenicity (section 'Classifier of immunogenic peptides'). The classifiers output a probability of being immunogenic which is used as the score for calculating the AUC on the 50 test sets of positives and negatives. To evaluate the AUC for the approach denoted as 'diffRBM units (difference)' (*Figure 4*, *Figure 4—figure supplement 6*), we considered the score given, for each test sequence $\boldsymbol{\sigma}$, by the difference $\mathcal{L}^d(\boldsymbol{\sigma}) - \mathcal{L}^{d,neg}(\boldsymbol{\sigma})$. Given that the background model is the same, it can be seen from *Equations 8 and 10* that the score $\mathcal{L}^d(\boldsymbol{\sigma}) - \mathcal{L}^{d,neg}(\boldsymbol{\sigma})$ gives the same result as the difference of the full RBM scores $\mathcal{L}(\boldsymbol{\sigma}) - \mathcal{L}^{neg}(\boldsymbol{\sigma})$ (*Figure 4—figure supplement 4*).

We followed the same procedure to train and evaluate the models of T-cell response specificity (*Figure 6*, *Figure 6—figure supplement 1*, *Figure 6—figure supplement 2*, *Figure 6—figure supplement 4*). To test the T-cell specificity models' ability to identify new peptide-specific receptors, we performed an AUC-based assessment of predictive performance using, as positives, the receptors with the same peptide-specificity from the held-out test sets and, as negatives, a subset of generic receptors from the bulk repertoire, randomly drawn at each repetition with the same size as the positive test set and with no overlap with the $10^6$ sequences of the training set of the background model. To further check the robustness of our results with respect to the choice of the background dataset, we repeated the training and testing constructing the background model from a different set

of bulk-repertoire TCRβ sequences from healthy donors (the dataset from *Britanova et al., 2016*). All the performance metrics are almost unaffected by this change of the background dataset (*Figure 6—figure supplement 2*). The average AUC attained by the diffRBM units across the 4 peptide-specific models is $0.83 \pm 0.01$ with the background dataset from *Emerson et al., 2017* and $0.84 \pm 0.01$ with the background dataset from *Britanova et al., 2016* (uncertainties are estimated over several training-test partitions). All the approaches (diffRBM and alternatives versions, SONIA, $k$-NN, NetTCR-2.0, TCRex) are trained and tested on 50 independent random partitions of both positives and negatives into training and test sets, and the performance shown in *Figure 6*, *Figure 6—figure supplement 1*, *Figure 6—figure supplement 2*, *Figure 6—figure supplement 4* is the average AUC over these 50 partitions. As negative set to train the supervised approaches ($k$-NN, NetTCR-2.0, TCRex) we took, for each training repetition, a subset of the bulk-repertoire dataset from *Emerson et al., 2017* with the same size as the positive training set.

## Contact prediction

### Definition of single-site factors

Given the TCR-pMHC structures retrieved from PDB and the estimated peptide-TCR and peptide-HLA contact sites (section 'Crystallographic structures from PDB'), we assessed whether differential models can predict contact positions. We defined single-site factors $T_i$ from the models' parameters to be evaluated on each sequence $\boldsymbol{\sigma}$ as:

$$T_i(\sigma_i) = g_i^{\mathrm{d}}(\sigma_i) + \sum_{\mu'=1}^{M^{\mathrm{d}}} w_{i\mu'}^{\mathrm{d}}(\sigma_i) \langle h_{\mu'}^{\mathrm{d}} | \boldsymbol{\sigma} \rangle \tag{11}$$

where the average over the differential hidden units $\langle h_{\mu'}^{\mathrm{d}} | \boldsymbol{\sigma} \rangle$ is estimated from a distribution conditional on the sequence $\boldsymbol{\sigma}$ that is $\sim \exp\left(-\mathcal{U}_{\mu'}^{\mathrm{d}}(h_{\mu'}^{\mathrm{d}}) + h_{\mu'}^{\mathrm{d}} \sum_i w_{i\mu'}^{\mathrm{d}}(\sigma_i)\right)$. As clear from the definition (*Equation 11*), single-site factors measure generally whether the amino acid $\sigma_i$ at position $i$, in the sequence context provided by $\boldsymbol{\sigma}$, has high probability to occur among selected sequences, for instance among HLA-specific immunogenic peptides, hence we have used them to quantify residue-specific contributions to immunogenicity (*Figure 3*).

For the prediction of contacts, the sequence $\boldsymbol{\sigma}$ in (*Equation 11*) is represented by either peptides or by receptors' CDR3β sequences when we predict, respectively, the peptide sites in contact with the TCR through the models of immunogenicity (*Figure 2C–D*) and the CDR3β sites in contact with peptides through the models of epitope specificity (*Figure 5B–C*). For each peptide, we use the immunogenicity model corresponding to its HLA type and for each CDR3β we use the model corresponding to its epitope specificity. Since we are interested in a prediction at the level of residues, the models of epitope specificity used here are defined only on the CDR3β amino acid sequence (disregarding the V and J identity). Given the set of $T_i(\sigma_i)$ for each sequence position $i$, we rank them according to their magnitude and we take the top ranking positions as the model's prediction on contacts for the sequence $\boldsymbol{\sigma}$. In the case of CDR3β sequences, we consider only non-gap positions for such ranking.

### Prediction assessment via the Positive Predictive Value

Given the models' predictions of putative contact sites, we assessed their quality by estimating the Positive Predictive Value (PPV, *Figure 2D*, *Figure 5C*, *Figure 2—figure supplement 3*, *Figure 2—figure supplement 4*, *Figure 5—figure supplement 3*). The PPV for sequence $\boldsymbol{\sigma}$ at the ranked position $p$ ($\mathrm{PPV}_{\boldsymbol{\sigma}}^p$) is given by the number of top $p$ ranked positions that is included among the contact positions of $\boldsymbol{\sigma}$ (true positives), divided by $p$ or by number of contacts when this is lower than $p$ (all the positives). $\mathrm{PPV}_{\boldsymbol{\sigma}}^p$ hence hits 1 when $p$ is equal to the full length of sequence $\boldsymbol{\sigma}$. For a given $\mathrm{PPV}_{\boldsymbol{\sigma}}^p$, the associated random expectation corresponds to drawing uniformly at random $p$ positions and using them to predict the contact positions of $\boldsymbol{\sigma}$. The summary values reported in *Figure 2D* and *Figure 5C* correspond to the average of $\mathrm{PPV}_{\boldsymbol{\sigma}}^p$ over all the sequences $\boldsymbol{\sigma}$ under consideration (respectively, peptides and CDR3β) as a function of the number $p$ of ranked sequence positions. To check that the average PPV values obtained are due to the predictive power of the diffRBM model, we performed a statistical hypothesis test based on the binomial distribution: we define as 'success'

the correct prediction of a contact using the diffRBM top ranked position and, using as null model a binomial distribution, we tested the null hypothesis that the probability of a success ($s$) is simply given by the average fraction of contact positions per sequence. We calculated the p-value of the hypothesis test as the probability, under the null model, of obtaining the number of successes corresponding to the diffRBM PPV value for the top ranked position out of a number of trials given by the number of sequences tested. We obtain statistically significant p-values for both the immunogenicity model (s=0.48, p-value = $7.7 \times 10^{-5}$) and the TCR specificity model (s=0.20, p-value = $1.6 \times 10^{-5}$). To correct for the over-representation among the available resolved structures of certain sequences (the same peptide and its one-point mutants in contact with different TCRs, sets of highly similar CDR3βs specific to the Influenza peptide GILGFVFTL), we calculate the average $PPV^p$ at each position $p$ as a weighted average:

$$PPV^p = \frac{1}{R_{\text{eff}}} \sum_{r=1}^{R} \frac{1}{n_{\boldsymbol{\sigma}^r}} PPV^p_{\boldsymbol{\sigma}^r} \tag{12}$$

where we reweight the contribution of each sequence to the PPV by a factor $1/n_{\boldsymbol{\sigma}}$, taking $n_{\boldsymbol{\sigma}}$ as the number of sequences that are equal to or one mutation away from $\boldsymbol{\sigma}$. In *Equation 12* we have denoted by $R$ the total number of entries under consideration and by $R_{\text{eff}}$ their effective number obtained as $R_{\text{eff}} = \sum_{r=1}^{R} 1/n_{\boldsymbol{\sigma}^r}$. Retaining only unique combinations of sequence and contact positions at the chosen distance cutoffs (4 Å for peptide-TCR contacts, 3.5 Å for peptide-HLA contacts), the number of structures considered for: (*i*) the prediction of contacts with the TCR along the peptide is $R = 46$ (4 for HLA-B*35:01, 41 for HLA-A*02:01, 1 for HLA-B*07:02); (*ii*) the prediction of contacts with the HLA along the peptide is $R = 53$ (5 for HLA-B*35:01, 47 for HLA-A*02:01, 1 for HLA-B*07:02); (*iii*) the prediction of contacts with the peptide along the CDR3β is $R = 12$ (2 for YLQPRTFLL, 3 for NLVPM-VATV, 1 for GLCTLVAML, 6 for GILGFVFTL). The corresponding effective numbers are $R_{\text{eff}} = 22.7$ for (*i*) and (*ii*), and $R_{\text{eff}} = 10.3$ for (*iii*). In *Figure 2—figure supplement 3A–B*, *Figure 2—figure supplement 4A–B*, *Figure 5—figure supplement 3A–B* we report the comparison of the average reweighted PPV (*Equation 12*) to the average PPV calculated without reweighting ($PPV^p = \frac{1}{R} \sum_{r=1}^{R} PPV^p_{\boldsymbol{\sigma}^r}$), showing that the reweighting does not affect the ranking of performance between different methods.

## Mutation costs

The experiments of *Łuksza et al., 2022* on how TCRs cross-react between the NLVPMVATV peptide and its mutants consisted of the following steps: the wild-type peptide (WT) was mutated to every amino acid at every position to obtain 171 mutants (MT); for each MT, its concentration was varied across a 10,000-fold range and the degree activation of 3 WT-specific TCRs was monitored as relative percentage of CD137 expression to determine the TCR cross-reactivity:

$$\text{TCR cross-reactivity} = \log \frac{EC_{50}^{MT}}{EC_{50}^{WT}} \tag{13}$$

from the WT and MT half maximal Effective Concentration $EC_{50}^{WT}$ and $EC_{50}^{MT}$ (both measured in $\mu g/ml$). We took these reported values of TCR cross-reactivity as the experimental mutation costs for each TCR/mutation pair for non-lethal mutations. We defined 'lethal' the mutations that were associated to a formally infinite $EC_{50}^{MT}$ in a given TCR context (*i.e.* TCR response could not be recovered even at the highest concentrations).

For a mutation in sequence $\boldsymbol{\sigma}$ at position $i$ changing $\sigma_i$ to $\sigma_i'$, we estimated the model prediction of the mutation cost as:

$$\text{Predicted mutation cost} = T_i(\sigma_i) - T_i(\sigma_i') \tag{14}$$

where we took $T_i$ as the single-site factors (*Equation 11*) for lethal mutations (*Figure 3E*) and of background RBM (*Equation 22* in Supporting Materials and methods) for non-lethal mutations (*Figure 3G*).

To assess whether the distribution towards positive values for lethal mutations is significantly higher than the expectation for generic, non immunogenicity-impacting mutations, we estimated the mutation cost distribution of a 'control' case (see *Figure 3E*) where we drew at random 3000 HLA-A*02:01-presented peptides from the background dataset, and we calculated the costs of all possible amino

acid substitutions at each position. The p-value for the difference in these distributions was estimated by the Mann-Whitney U test.

## Acknowledgements

We thank Julien Racle, Nathanael Spisak, Francesco Camaglia, Marta Łuksza, Christopher Thorpe, Morten Nielsen and Vadim Karnaukhov for discussions. We are particularly grateful to Giulio Isacchini for help with the data and software. This research was supported by the ANR-17-CE30-0021-01 RBMPro, ANR-19-CE30-0021-01 Decrypted, ANR-19-CE45-0018 RESP-REP, Prodigen grants from the Agence Nationale de la Recherche and by the European Research Council COG 724208. ADG acknowledges funding from the European Union's Horizon 2020 research and innovation programme under the Marie Sklodowska-Curie grant agreement No 101026293.

## Additional information

### Competing interests

Aleksandra M Walczak: Senior editor, eLife. The other authors declare that no competing interests exist.

### Funding

| Funder | Grant reference number | Author |
|---|---|---|
| Agence Nationale de la Recherche | RBMPro CE30-0021-01 | Andrea Di Gioacchino Jorge Fernandez-de-Cossio-Diaz Simona Cocco Rémi Monasson |
| Agence Nationale de la Recherche | ProDiGen CE45-2023 | Andrea Di Gioacchino Jorge Fernandez-de-Cossio-Diaz Simona Cocco Rémi Monasson |
| Agence Nationale de la Recherche | Decrypted CE30-0021-01 | Andrea Di Gioacchino Jorge Fernandez-de-Cossio-Diaz Simona Cocco Rémi Monasson |
| Agence Nationale de la Recherche | RESP-REP CE45-0018 | Barbara Bravi Aleksandra M Walczak Thierry Mora |
| European Research Council | COG 724208 | Barbara Bravi Aleksandra M Walczak Thierry Mora |
| HORIZON EUROPE Marie Sklodowska-Curie Actions | 101026293 | Andrea Di Gioacchino |

The funders had no role in study design, data collection and interpretation, or the decision to submit the work for publication.

### Author contributions

Barbara Bravi, Conceptualization, Data curation, Software, Formal analysis, Validation, Investigation, Visualization, Methodology, Writing - original draft; Andrea Di Gioacchino, Jorge Fernandez-de-Cossio-Diaz, Conceptualization, Software, Formal analysis, Validation, Investigation, Methodology, Writing – review and editing; Aleksandra M Walczak, Thierry Mora, Simona Cocco, Rémi Monasson, Conceptualization, Formal analysis, Supervision, Funding acquisition, Investigation, Methodology, Writing – review and editing

## Author ORCIDs
Barbara Bravi http://orcid.org/0000-0003-4860-7584
Andrea Di Gioacchino https://orcid.org/0000-0002-6085-7589
Jorge Fernandez-de-Cossio-Diaz http://orcid.org/0000-0002-4476-805X
Aleksandra M Walczak https://orcid.org/0000-0002-2686-5702
Thierry Mora https://orcid.org/0000-0002-5456-9361
Simona Cocco http://orcid.org/0000-0002-1852-7789
Rémi Monasson http://orcid.org/0000-0002-4459-0204

## Decision letter and Author response
Decision letter https://doi.org/10.7554/eLife.85126.sa1
Author response https://doi.org/10.7554/eLife.85126.sa2

# Additional files

## Supplementary files
• MDAR checklist

## Data availability
The current manuscript is a computational study, so no data have been generated for this manuscript. The data used are downloaded from public databases. The pre-processed data, the results of the analysis, the codes to train and evaluate the models as well as the trained models are all available at the github page https://github.com/bravib/diffRBM_immunogenicity_TCRspecificity (copy archived at *Bravi, 2023*).

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

# Appendix 1

## Comparison of performance with existing tools

### DiffRBM model of antigen immunogenicity

We compared the discrimination performance of the diffRBM model of antigen immunogenicity to two established methods for immunogenicity prediction, the IEDB immunogenicity tool and PRIME (*Schmidt et al., 2021*). Since these methods are not designed to be re-trained on custom datasets such as ours, the comparison cannot be fully consistent in terms of using the *same* training and test sets, as done for all the methods in *Figure 4*. The IEDB tool for immunogenicity prediction is based on the model by *Calis et al., 2013* and can be downloaded from http://tools.iedb.org/immunogenicity/. By applying it to our set of immunogenic and non-immunogenic peptides, we obtained AUC = 0.54 for the HLA-A*02:01-specific peptides, AUC = 0.60 for HLA-B*07:02 and AUC = 0.57 for HLA-B*35:01, which are all lower than the diffRBM average AUC values (respectively 0.66, 0.65, 0.67, see *Figure 4*).

PRIME (*Schmidt et al., 2021*) was downloaded from https://github.com/GfellerLab/PRIME (*Gfeller, 2022*). Since it was not possible to re-train PRIME on our datasets for a fair comparison of performance, we simply evaluated it on the set of immunogenic and non-immunogenic peptides we collected and we obtained a discrimination performance with AUC = 0.53 for peptides specific to HLA-A*02:01, AUC = 0.45 for HLA-B*07:02 and AUC = 0.56 for HLA-B*35:01. Since we set out to predict immunogenicity conditioned on binding to a given HLA allele, PRIME was run in the mono-allelic mode (*e.g.* with option *-a A0201* for the case of HLA-A*02:01); in general, however, different results are obtained by adding more alleles, where then the best presenting allele according to the predictor is taken. In summary, hence, we found poorer performance compared to the diffRBM units, despite not having excluded from our test sets the immunogenic peptides that are either in PRIME's or in the IEDB tool's training sets. Differences in the peptide datasets used may contribute to explain this result. For instance, our dataset consists of epitopes with experimentally validated positive T-cell responses from IEDB, mainly of microbial origin, while PRIME's training set (*Schmidt et al., 2021*) was constructed in such a way as to contain a high proportion of neoantigens.

As an additional point of comparison, (*Riley et al., 2019*) propose a neural network trained in a supervised way to classify immunogenic against non-immunogenic peptides using sequence as well as structural features of the peptide/HLA complex. When they check the performance of a peptide sequence-only predictor (using nonamer HLA-A*02:01-restricted peptides), they obtain AUC = 0.61 on the training set and AUC = 0.50 on the test. This further confirms that the performance of our diffRBM approach, despite not being directly trained as a discriminator of immunogenic *vs* non-immunogenic peptides, compares favorably to existing sequence-based tools of prediction.

### DiffRBM model of TCR specificity

To compare diffRBM to state-of-the-art predictors, we chose to take into consideration TCRex (*Gielis et al., 2019*) and NetTCR-2.0 (*Montemurro et al., 2021*), which were found to attain the best performance at predicting TCR specificity in a recent benchmark of different methods (*Meysman et al., 2022*). TCRex is a random forest classifier of TCRs by epitope specificity. The most recent version is publicly accessible at https://github.com/bittremieux/TCR-classifier (*De Neuter and Bittremieux, 2020*), and can be used as described in *De Neuter et al., 2018*. As a modification to the pipeline described in *De Neuter et al., 2018*, we decided to skip the feature selection step (originally performed through the Boruta algorithm) after checking that the performance of the classification task is not affected by this choice in our case. NetTCR belongs to a family of methods (*Springer et al., 2020*; *Weber et al., 2021*; *Milighetti et al., 2021*) trained on TCR-antigen pairs to predict a score of binding, but can also be applied to the prediction of whether an unseen TCR is likely to be specific to a given peptide target. In particular, NetTCR-2.0 (*Montemurro et al., 2021*) is a convolutional neural network method trained from sets of pairs of peptides and cognate TCR sequences ($\beta$ chain only or $\alpha + \beta$ chains). To make a fair comparison of performance, we re-trained these methods on our training sets and evaluated the AUC of discrimination of specific receptors on our test sets, considering their version for the β-chain only (section 'Classification performance').

To re-train and evaluate NetTCR-2.0 with our datasets, we replaced the sets of CDR3β specific to the 4 epitopes of interest with our VDJdb-derived datasets. We kept the other NetTCR training data

(3 additional peptides with their specific CDR3β as well as their control data). Re-training NetTCR-2.0 with only the 4 epitope-specific sets of CDR3β we considered gave AUCs in the same range as for the case with 7 peptides. Our training of NetTCR-2.0 uses the same default settings as **Montemurro et al., 2021**. While the performance of TCRex is the same as diffRBM (both with an average AUC across peptides of $0.83 \pm 0.01$, see **Figure 6B–E**), we observed a moderately lower performance of NetTCR-2.0 (AUC = $0.73 \pm 0.01$), which could be partially due to the fact that the CDR3β version of NetTCR-2.0 does not take as input the V and J genes. To check the role of including information on V and J, we also re-trained diffRBM using only the CDR3β amino acid sequence (**Figure 6— figure supplement 4**). The performance of the diffRBM units is reduced (AUC = $0.79 \pm 0.01$), but still higher than NetTCR-2.0. To test the susceptibility to predict false positives of diffRBM compared to state-of-the-art methods, we calculated also the Average Precision score (AP) and we obtained that diffRBM performs better or as well as the other methods across the 4 epitopes (GILGFVFTL: diffRBM has an average AP = 0.87 across the 50 training/test partitions, TCRex AP = 0.85, NetTCR-2.0 = 0.77; NLVPMVATV: diffRBM AP = 0.79, TCRex AP = 0.78, NetTCR-2.0 = 0.67; GLCTLVAML: diffRBM AP = 0.82, TCRex AP = 0.81, NetTCR-2.0 = 0.70; YLQPRTFLL: diffRBM AP = 0.99, TCRex AP = 0.99, NetTCR-2.0 = 0.89). Note it is at the task of distinguishing specific TCRs from background data that diffRBM compares in this way to state-of-the-art methods. The performance by diffRBM decreases when we consider as negatives receptors with a different antigen specificity (**Meysman et al., 2022**), because the differential units can learn features shared between antigen-specific sequences (*e.g.*, being expressed by CD8+T cells). More fine-tuned choices of the background (*e.g.*, with only CD8+T cell receptors) would be needed to improve performance for this particular task.

## Appendix 2

### Details on training diffRBM

Let us define a compact notation $\mathbf{\Psi}^{\mathrm{d}}$ for the set of parameters associated to the differential units, $\mathbf{\Psi}^{\mathrm{d}} = \{g_i^{\mathrm{d}}(\sigma_i), w_{i\mu'}^{\mathrm{d}}(\sigma_i), \xi_{\mu'}^{\mathrm{d}}\}$, where $\xi_{\mu'}^{\mathrm{d}}$ is a shorthand for the parameters specifying the shape of $\mathcal{U}_{\mu'}^{\mathrm{d}}$ (see *Equation 5* in Materials and methods). In all our implementations, both $\mathcal{U}_{\mu}^{\mathrm{b}}$ and $\mathcal{U}_{\mu'}^{\mathrm{d}}$ are set to dReLU potentials, a choice that has been shown to confer high expressivity (*Tubiana et al., 2019*). The rules to infer $\mathbf{\Psi}^{\mathrm{d}}$ are given by gradient-ascent equations to maximize the likelihood of the post-selection data $\mathcal{D}^{\mathrm{s}}$ under the full RBM model (*Equation 1*):

$$\frac{\partial}{\partial \mathbf{\Psi}^{\mathrm{d}}} \langle \log P(\boldsymbol{\sigma}) \rangle_{\mathcal{D}^{\mathrm{s}}} = \left\langle \frac{\partial}{\partial \mathbf{\Psi}^{\mathrm{d}}} \mathcal{H}(\boldsymbol{\sigma}) \right\rangle_m - \left\langle \frac{\partial}{\partial \mathbf{\Psi}^{\mathrm{d}}} \mathcal{H}(\boldsymbol{\sigma}) \right\rangle_{\mathcal{D}^{\mathrm{s}}} \tag{15}$$

where $\langle\rangle_m$ indicates the average under the full RBM model, $\langle f \rangle_m = \sum_{\boldsymbol{\sigma}} \frac{1}{Z} f(\boldsymbol{\sigma}) e^{-\mathcal{H}^{\mathrm{b}}(\boldsymbol{\sigma}) - \mathcal{H}^{\mathrm{d}}(\boldsymbol{\sigma})}$. Since the parameters $\mathbf{\Psi}^{\mathrm{d}}$ to learn appear only in $\mathcal{H}^{\mathrm{d}}$, one has $\partial \mathcal{H}(\boldsymbol{\sigma})/\partial \mathbf{\Psi}^{\mathrm{d}} = \partial \mathcal{H}^{\mathrm{d}}(\boldsymbol{\sigma})/\partial \mathbf{\Psi}^{\mathrm{d}}$; the contribution of the background $\mathcal{H}^{\mathrm{b}}$ enters therefore only in the estimation of the average $\langle\rangle_m$, which requires, at each step of the training, to sample configurations with the probability (*Equation 1*).

Regularization terms are typically added during the training to control the values of the inferred parameters. Here we used a $L_1$-type regularization over the weights, which enforces sparsity to prevent overfitting (see *Tubiana et al., 2019* for details); following the convention in *Tubiana et al., 2019* we denote by $\lambda_2^1$ the coefficient setting the magnitude of such regularization term. Additional $L_2$ regularization was imposed on the weigths, with amplitude $1/|\mathcal{D}^{\mathrm{b}}|$ for background models and $1/|\mathcal{D}^{\mathrm{s}}|$ for differential models (*Cocco et al., 2018*).

## Appendix 3

### Alternative approaches tested

We compared diffRBM performance either to alternative architectures for differential models (diffRBM linear, PWM-based approach, SONIA) or to fully supervised, background-free models (classifiers).

### DiffRBM linear

DiffRBM linear is a diffRBM architecture where the differential part is specified by single-site fields $g_i^{\mathrm{d}}(\sigma_i)$ (without the addition of weights $w_{i\mu'}^{\mathrm{d}}(\sigma_i)$).

### PWM-based approach

We considered a differential model based entirely on single-site amino frequency, where both the background model and the differential part are specified only by single-site fields ($g_i^{\mathrm{b}}(\sigma_i)$ and $g_i^{\mathrm{d}}(\sigma_i)$ respectively). The equations for learning $g_i^{\mathrm{b}}(\sigma_i)$ and $g_i^{\mathrm{d}}(\sigma_i)$ are:

$$\frac{e^{g_i^{\mathrm{b}}(\sigma_i)}}{\sum_{\sigma_i} e^{g_i^{\mathrm{b}}(\sigma_i)}} + \lambda_2 g_i^{\mathrm{b}}(\sigma_i) = f_i^{\mathrm{b}}(\sigma_i) \tag{16}$$

$$\frac{e^{(g_i^{\mathrm{b}}(\sigma_i)+g_i^{\mathrm{d}}(\sigma_i))}}{\sum_{\sigma_i} e^{(g_i^{\mathrm{b}}(\sigma_i)+g_i^{\mathrm{d}}(\sigma_i))}} + \lambda_2(g_i^{\mathrm{b}}(\sigma_i) + g_i^{\mathrm{d}}(\sigma_i)) = f_i(\sigma_i) \tag{17}$$

where $f_i^{\mathrm{b}}(\sigma_i)$ is the frequency of amino acid $\sigma_i$ at position $i$ in the background dataset $\mathcal{D}^{\mathrm{b}}$, $f_i(\sigma_i)$ is the frequency of amino acid $\sigma_i$ at position $i$ in the dataset $\mathcal{D}^{\mathrm{s}}$ and $i = 1, ..., N$. We use the same $L_2$ regularization penalty that we chosen for the RBM fields ($\lambda_2 = 1/|\mathcal{D}^{\mathrm{b}}|$ for background models and $\lambda_2 = 1/|\mathcal{D}^{\mathrm{s}}|$ for differential part). This approach is equivalent to learning from the background and selected datasets Position Weight Matrices (PWMs), probabilistic models of amino acid usage that treat all sequence positions as independent. By defining $P_i^{\mathrm{b}}(\sigma_i) = e^{g_i^{\mathrm{b}}(\sigma_i)}/\sum_{\sigma_i} e^{g_i^{\mathrm{b}}(\sigma_i)}$ and $P_i(\sigma_i) = e^{(g_i^{\mathrm{b}}(\sigma_i)+g_i^{\mathrm{d}}(\sigma_i))}/\sum_{\sigma_i} e^{(g_i^{\mathrm{b}}(\sigma_i)+g_i^{\mathrm{d}}(\sigma_i))}$ and given the independent-site assumption of PWMs, the background data PWM probability and the selected data PWM probability are respectively recovered as $P^{\mathrm{b}}(\boldsymbol{\sigma}) = \prod_i P_i^{\mathrm{b}}(\sigma_i)$ and $P(\boldsymbol{\sigma}) = \prod_i P_i(\sigma_i)$. Since $P_i^{\mathrm{b}}(\sigma_i)$ and $P_i(\sigma_i)$ are learnt to closely reproduce the frequencies $f_i^{\mathrm{b}}(\sigma_i)$ and $f_i(\sigma_i)$, their predictions are the ones that are inferable from single-site amino acid frequency alone in the respective training datasets (hence in *Figure 2*, *Figure 5*, *Figure 2— figure supplement 3*, *Figure 2—figure supplement 4*, *Figure 5—figure supplement 3* we refer to such predictions by 'Amino-acid frequency'). For a sequence $\boldsymbol{\sigma}$, the score based on the differential part is then simply given by the log-likelihood ratio:

$$\mathcal{L}^{\mathrm{d}}(\boldsymbol{\sigma}) = \sum_i \log \frac{P_i(\sigma_i)}{P_i^{\mathrm{b}}(\sigma_i)} \tag{18}$$

measuring the site-specific enrichment in amino-acid usage in selected data compared to the background.

### SONIA

We implemented a SONIA model using the software package from *Isacchini et al., 2021* and setting the option *deep = False*. For consistency with the background of diffRBM, we first learn a background model from $10^6$ randomly assembled from the universal donor repertoire from *Emerson et al., 2017*, next we learn a model from epitope-specific repertoires and we take, as SONIA score, the difference between the scores assigned by these two models. The $L_2$-type regularization strength is set to the default value (zero); higher regularizations applied to one of our test sets did not provide visible improvement of the log-likelihood.

### Classifier of immunogenic peptides

As a term of comparison for the immunogenicity model, we implemented first a linear (logistic) classifier (see Model C1 in *Figure 4—figure supplement 5C*), which we trained by minimization of a binary cross-entropy loss. To find an optimal classifier, we also searched among different neural-network architectures trained to discriminate the immunogenic and non-immunogenic data retrieved from

IEDB (section 'Sequence datasets for the immunogenicity model'). We used the same 50 validation sets held out from the training set of immunogenic, HLA-A*02:01-presented peptides utilized for the model selection of diffRBM (section 'DiffRBM training and model selection'). In an analogous way, we randomly partitioned the set of non-immunogenic, HLA-A*02:01-presented peptides into training/test sets (with respectively 80%/20% of the data), repeating the partition 50 times, and we further partitioned the training sets to hold out at each repetition a validation set with 20% of sequences. We trained the models varying the number and width of hidden layers (*Figure 4—figure supplement 5C*), as well as the weight decay. The 'optimal classifier' performance for HLA-A*02:01 (*Figure 4*) is obtained by selecting for each partition the best performing architecture on the validation set and evaluating its AUC on the test set. Next, we identified the architecture giving the maximal average AUC of discrimination between the immunogenic and non-immunogenic peptides in the validation sets (Model C8 in *Figure 4—figure supplement 5C*) and we used it to estimate the 'optimal classifier' average AUC for the HLA-B*07:02 and HLA-B*35:01 models (*Figure 4C–D*). Our numerical implementation is based on the PyTorch library. Training is performed in mini-batches of 64 sequences, by the AdamW optimizer with weight decay (*Loshchilov and Hutter, 2019*), learning rate 0.001, for 500 epochs.

## $k$-NN based classifier

In *Weber et al., 2021* a $k$-Nearest Neighbours ($k$-NN) classifier served as a baseline method to determine the predictive power achievable from TCR sequence similarity alone and, despite its simplicity, it was found to outperform some existing methods. Thus, following *Weber et al., 2021*, we built a $k$-Nearest Neighbours ($k$-NN) based classifier to distinguish TCRs recognizing a specific antigen from generic ones. The method works by computing the Levenshtein distance of the CDR3β under analysis with respect to a set of positive examples (training set of antigen-specific CDR3β) and to a set of negative examples (training set of bulk repertoire CDR3β). In the computation of the Levenshtein distance, the V and J segments are used as well (different V or J segments increase the distance by 1). Next the average distances from the $k$ nearest neighbours are computed for both the positive and negative examples, and a score is computed as the difference between the two average distances.

The only parameter of this model, $k$, has been fixed for the largest dataset of positive examples (TCRs reactive to the NLVPMVATV peptide) by cross-validation. We split the positive and negative data in 50 independent training and test sets (with respectively 80% and 20% of the sequences); from each training set, a portion of 20% of its data is held out as validation set. Next $k$, for each partition, is fixed by maximizing the model performance on the validation set, and the model with the best $k$ is evaluated on the test set. For the other datasets of positive examples, which are smaller, we used $k = 26$, obtained as the $k$ for which the average AUC on the validation set is maximal (*Figure 6—figure supplement 3*). The performance of the $k$-NN based classifier was evaluated as described in section 'Classification performance'.

## Single-site factors for the alternative approaches

As a term of comparison of diffRBM single-site factors (*Equation 11* in Materials and methods), we considered single-site factors from the diffRBM linear model $T_i(\sigma_i) = g_i^{\mathrm{d}}(\sigma_i)$ (*Figure 2—figure supplement 3A–B*, *Figure 5—figure supplement 3A–B*), and from models which assume all positions independent (section 'PWM-based approach'). Here the single-site factors reduce to the log-likelihood ratio:

$$T_i(\sigma_i) = \log \frac{P_i(\sigma_i)}{P_i^{\mathrm{b}}(\sigma_i)} \qquad (19)$$

where that $P_i(\sigma_i)$ and $P_i^{\mathrm{b}}(\sigma_i)$ are given by PWMs (section 'PWM-based approach') and hence represent position-specific amino acid frequencies. In the case of the immunogenicity model, the prediction by *Equation 19* simply describes the log ratio between the frequency of amino acids in immunogenic peptides relative to the amino acid frequency of all presented peptides from the background dataset (we refer to it as 'AA frequency ratio (to all presented)', see *Figure 2C–D*); in the case of the TCR specificity model *Equation 19* gives the log ratio between the frequency of amino acids in peptide-specific CDR3β repertoires relative to the amino acid frequency in bulk repertoire (we refer to it as 'AA frequency ratio (to bulk)', see *Figure 5B–C*). In both cases the prediction indicated as 'Amino acid frequency' is obtained by estimating and ranking the single-site factors:

$$T_i(\sigma_i) = \log P_i(\sigma_i) \tag{20}$$

In the case of the immunogenicity model, we have also learnt an independent-site model from non-immunogenic data (that we call $P^{\mathrm{NI}}(\sigma_i)$) and we have looked at the single-site factors:

$$T_i(\sigma_i) = \log \frac{P_i(\sigma_i)}{P_i^{\mathrm{NI}}(\sigma_i)} \tag{21}$$

which represents a log ratio of amino acid frequency in immunogenic data relative to non-immunogenic data (this prediction is labeled 'AA frequency ratio (to non-immunogenic)' in *Figure 2C–D* and *Figure 3B*). To identify the sites of binding to the HLA, we took first single-site factors simply capturing amino acid usage in presented peptides defined as:

$$T_i(\sigma_i) = \log P_i^{\mathrm{b}}(\sigma_i) \tag{22}$$

and labeled as 'Amino Acid Frequency' in *Figure 2B* and *Figure 2—figure supplement 4*. Next we looked at the parameters of background RBM by defining single-site factors that correspond essentially to the model's log probability of residue $\sigma_i$ conditional on the rest of the residues:

$$T_i(\sigma_i) = g_i^{\mathrm{b}}(\sigma_i) + \sum_{\mu=1}^{M^{\mathrm{b}}} w_{i\mu}^{\mathrm{b}}(\sigma_i)\langle h_\mu | \boldsymbol{\sigma} \rangle \tag{23}$$

We also considered a less refined version including only the fields of the background RBM model (hence not accounting for the interaction with the rest of the sequence sites):

$$T_i(\sigma_i) = g_i^{\mathrm{b}}(\sigma_i) \tag{24}$$

and we observed that this fields-only prediction performs slightly better at identifying the anchor sites of binding to the HLA compared to the full RBM prediction (*Figure 2—figure supplement 4A–B*). Indeed, the background RBM weights tend also to capture amino acid differences at the peptide's central positions, which are likely to reflect the existence of variable peptide conformations within the binding pocket (*Nguyen et al., 2021*). A case in point was discussed in *Bravi et al., 2021b*, where we found that the weights of a single-allele RBM presentation model captured different modes of binding across peptides of the same HLA specificity. This would help explain why fields alone perform slightly better as predictors of the primary anchor sites of binding to the HLA, and in *Figure 2B* we use the single-site factors from a fields-only RBM model as predictors. Anyway the prediction by a full RBM model (*Figure 2—figure supplement 4A–B*) exhibits only a minimal decrease in performance compared to the fields-only prediction, due to the weights-related effects being mainly second-order ones.

Finally, we have explored an alternative, more general measure of site-specific amino acid importance from the differential units:

$$\widetilde{T}_i(\sigma_i) = \mathcal{L}^{\mathrm{d}}(\boldsymbol{\sigma}) - \frac{1}{q}\sum_{\sigma_i'=1}^{q} \mathcal{L}^{\mathrm{d}}(\sigma_i'|\boldsymbol{\sigma}) \tag{25}$$

where $\mathcal{L}^{\mathrm{d}}(\sigma_i'|\boldsymbol{\sigma})$ is the differential units' score of the sequence $\boldsymbol{\sigma}$ where position $i$ has been mutated from $\sigma_i$ to $\sigma_i'$, $\mathcal{L}^{\mathrm{d}}$ is calculated using *Equation 10* in Materials and methods and $q$ is the number of values that $\sigma_i'$ can take. $\widetilde{T}_i(\sigma_i)$ can be hence seen as a measure of the importance of amino acid $\sigma_i$ at position $i$ obtained by comparing its contribution to the likelihood to the one of all other possible $q$ amino acids. The diffRBM single-site factors (*Equation 11* in Materials and methods) can be derived as a small-weight approximation of the more general definition (*Equation 25*), whose advantage is that per se does not depend on the RBM structure of the probability $P(\boldsymbol{\sigma})$. Overall, at the numerical level, the predictions of contacts obtained by ranking sites by *Equation 25* are comparable to the ones by *Equation 11*, see *Figure 2—figure supplement 3A–B*, *Figure 2—figure supplement 4A–B*, *Figure 5—figure supplement 3A–B*.

# Appendix 4

## Entropy estimation

The entropies of background RBM and full RBM (*Appendix 4—figure 1A–B*) were estimated as:

$$\text{Background RBM:} \quad S = -\sum_{\sigma} P^{\text{b}}(\sigma) \log P^{\text{b}}(\sigma) \qquad \text{Full RBM:} \quad S = -\sum_{\sigma} P(\sigma) \log P(\sigma) \qquad (26)$$

where $P^{\text{b}}(\sigma)$ and $P(\sigma)$ are given respectively by *Equations 1 and 2* in Materials and methods. The log is meant as a natural logarithm, hence the entropy values in *Appendix 4—figure 1A–B* are expressed in nats. To estimate the entropy of PWM-based approaches (*Appendix 4—figure 1C*), we used *Equation 26* with PWM probabilities for $P^{\text{b}}(\sigma)$ and $P(\sigma)$ (see section 'PWM-based approach', Appendix 3).

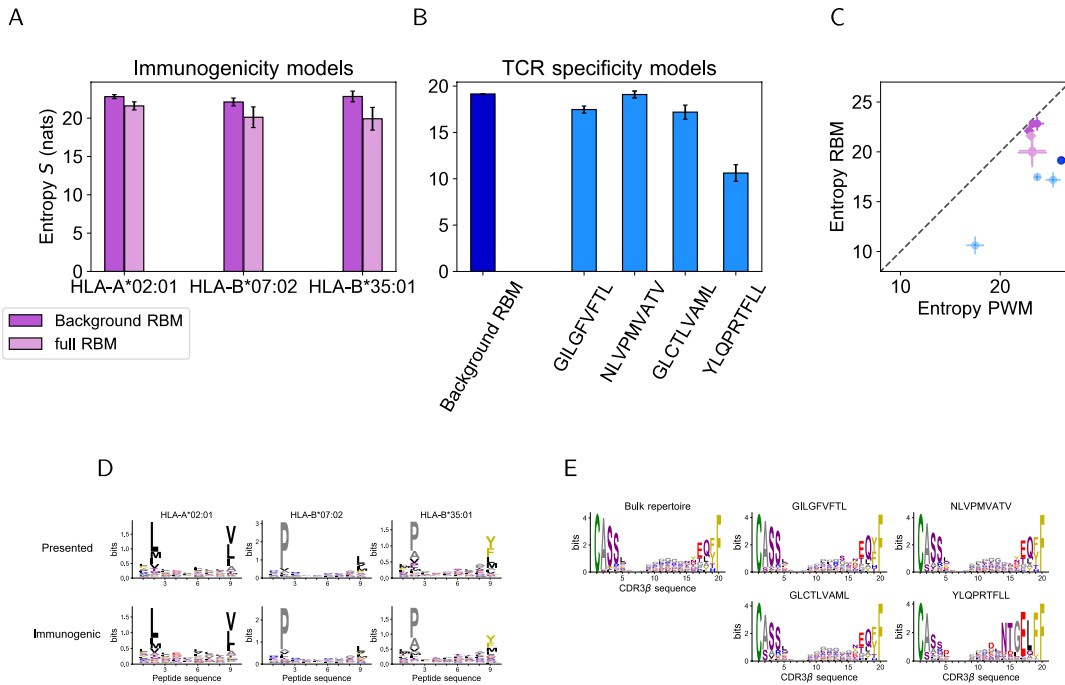

**Appendix 4—figure 1.** Model-based entropy estimation. (**A**) Entropy (expressed in nats) of the space of HLA-specific presented antigens (evaluated by background RBM) and of HLA-specific immunogenic antigens (evaluated by the full RBM) for the 3 HLAs. Error bars represent the sampling-related uncertainty on the estimated entropy and was calculated as in *Marchi et al., 2019*. (**B**) The entropy of the background dataset (CDR3 bulk repertoire) obtained from background RBM is compared to the entropy of epitope-specific CDR3 obtained from the full RBM models of TCR specificity to GILGFVFTL, NLVPMVATV, GLCTLVAML and YLQPRTFLL. (**C**) The entropies calculated from background RBM and the full RBM plotted in A and B is lower than the one estimated from independent-site models of the same data (Entropy PWM), because RBM models can account for correlations between sequence sites, hence for additional constraints on sequence diversity. (Colors are the same as in **A, B**). The entropy values from RBM and PWM models show a highly correlated trend across datasets, reflecting their different degree of heterogeneity in amino acid composition, as shown by the sequence logos of peptide (**D**) and CDR3 (**E**) data we considered.

