## [Editor Report]

In this important work, the authors present a sequence-based approach using transfer learning and Restricted Boltzmann Machines to predict antigen immunogenicity and specificity. The evidence and methodology are compelling. This work should be of interest to immunologists, computational biologists, and biophysicists.

---

## [Decision Letter]

**Decision letter after peer review:**

Thank you for submitting your article "Learning the differences: a transfer-learning approach to predict antigen immunogenicity and T-cell receptor specificity" for consideration by *eLife*. Your article has been reviewed by 3 peer reviewers, and the evaluation has been overseen by a Reviewing Editor and Naama Barkai as the Senior Editor. The reviewers have opted to remain anonymous.

Essential revisions:

1) Please streamline the manuscript, which is very complete and rigorous, but also quite long and sometimes intricate. In particular:

– In the Discussion, please emphasize the main conclusions regarding the performance of the method, by concisely explaining for the general reader in what cases it is better than existing methods and what new insights it can provide for immunology (see also Reviewer #1 and Reviewer #3's comments about this).

– The various comparisons to other methods are definitely important and useful, but they sometimes obscure the main points of the paper. We advise to state the conclusions of the various comparisons to other methods in the main text, but to put some of the details of these analyses in Supplement (see also Reviewer #2's comment about this).

2) Please consider including comparisons to transfer learning methods, as advised by Reviewer #3. If feasible, comparisons to general transfer learning tools would be interesting and could strengthen the evidence for broad applicability. If these comparisons are not feasible, please explain why.

*Reviewer #2 (Recommendations for the authors):*

General Questions

I will provide here some general questions that if addressed could, in my opinion, help improve the clarity of this article.

1. I found the formulation aimed to disentangle the background from selected features really interesting. The way the learning is done is that both datasets are known a priori and then the learning happens sequentially. But what if the datasets where the two generic and specific features are mixed? How would the learning happen in this case? If DiffRBM cannot be used in this situation then it is ok, but it would be good to discuss it as a limitation.

2. The paper mentions that the learning happened in an HLA-specific approach. Although advantages and disadvantages are presented throughout the paper, it would be good to address this important distinction in the discussion.

3. Although I understand this article's main goal is to present a technical solution to specific phenomena in immunology, after reading the manuscript I felt that more specific direct applications should have been discussed. How can the scientific community, take this approach that has been extensively validated and compared and translate it into potential therapeutic or scientific applications?

4. When describing contact prediction and the ranking of peptide positions, the result seemed quite encouraging, however, when one deals with such a small dataset (6-9 positions) then statistical significance becomes a relevant question. Can the authors also discuss this in the presentation of the results? This might help prove a more realistic representation of the predictive qualities of this side of DiffRBM.

5. In Figure 3G the authors estimate the mutational cost of peptides compared to experimental measures. The correlation although significant seems low to me r=0.47. Can the authors comment if this is a limitation or if in fact, other methods cannot reach better correlations than the one presented using DiffRBM?

*Reviewer #3 (Recommendations for the authors):*

The authors propose a novel model architecture called diffRBM, which is based on the original RBM papers [Hinton, 2002, Hinton and Salakhutdinov, 2006], by adding separate background and differential units and a specific training procedure, as described in section 5.1, opening a possibility to use this model for the transfer of learning.

To be interesting to the wide readership of *eLife* this paper should show novelty both in terms of the biology it describes and the(AI transfer of learning) tools. Especially as the tool's novelty is emphasized by the authors.

While the paper is very long and detailed it is still a bit unclear to us why other than being novel in its AI methods it is better or more revealing than existing (also non-AI) methods in terms of our understanding of TCR immunology.

We also have the following questions about the methodology:

1. Since it is a novel transfer of learning method, we would like to see a general comparison of this method with other transfer of learning methods on standard benchmarks, not only on the immunological data. Authors claim in section 2 that "… it could be applied to any data that has some distinctive features compared to a much larger pool of data endowed with the baseline properties". We also believe that such a method could be useful to a broader community of researchers. Also, by performing a comparison on standard benchmarks, the novelty of diffRBM could be highlighted more clearly, since it will be certain whether this model consistently achieves state-of-the-art results or not.

We suggest two directions:

1.1 Performing a comparison to the classical transfer of learning and domain adaptation papers, that provide generally good transfer of learning methods, used by a broad research community. Such papers also talk about the transfer of learning benchmark datasets. While most of the datasets are about image data, text-based datasets, more closely resembling biological data, can be found as well in related papers.

An example classical paper:

Adversarial Discriminative Domain Adaptation, Eric Tzeng et. al

1.2. Performing a comparison to the more recent transfer of learning and domain adaptation papers. While it is up to the authors to analyze and define the scope of applicability of the proposed diffRBM model, we can suggest selecting some of the recent papers and benchmark datasets from the links: https://paperswithcode.com/task/domain-adaptation; https://paperswithcode.com/sota/domain-adaptation-on-visda2017. Same comment about closeness to biological data is applicable here as well: it may be necessary to find more specific standard datasets, closely resembling the biological data, if diffRBM is supposed to work well specifically on such data domains.

2. On the other hand, when applied to immunological data (that is described in the paper), we also would like to see more models from the classical and recent transfer of learning papers to be compared to. So if applicable, adding classical and recent transfer of learning models from the above links to the comparison would be beneficial.

Note: If in fact, diffRBM is not widely applicable beyond specific immunological/sequence data – it would be nice to see a detailed analysis as to what properties of the immunological data make diffRBM applicable to it, and not to other data.

---

## [Author Response]

Essential revisions:1) Please streamline the manuscript, which is very complete and rigorous, but also quite long and sometimes intricate. In particular:– In the Discussion, please emphasize the main conclusions regarding the performance of the method, by concisely explaining for the general reader in what cases it is better than existing methods and what new insights it can provide for immunology (see also Reviewer #1 and Reviewer #3's comments about this).

We have re-written several paragraphs in the discussion, addressing directly the advantages and limitations of our approach compared to other methods, explaining the type of insights and predictions they can provide, and mentioning a few related applications in the biomedical context. We have also divided the discussion into small sections. Following Reviewer 1’s comments, we now make clear that diffRBM, contrary to most existing methods, needs only positive examples for its training, and is thus ‘semi-supervised’.

– The various comparisons to other methods are definitely important and useful, but they sometimes obscure the main points of the paper. We advise to state the conclusions of the various comparisons to other methods in the main text, but to put some of the details of these analyses in Supplement (see also Reviewer #2's comment about this).

We have significantly shortened the sections devoted to the comparison to other methods, and moved the details to the Appendix (‘Appendix 1 – Comparison of performance with existing tools’). This SI section also includes details on our implementation of TCRex and NetTCR. We have also moved the summary on state-of-the-art methods to the Introduction (lines 43-62) to improve the readability of the Results section.

We have streamlined the sections ‘Validation of model predictions against TCR-pMHC structures’ and ‘DiffRBM encodes molecular features of immunogenicity’. We have also shortened the Methods, adding Appendix 2, 3, 4 where we have moved details on diffRBM training, estimation of single-site factors and entropies and the description of the alternative approaches tested.

While a few paragraphs have been rephrased to be more concise, or moved to the Appendix as we said, we have highlighted in a different colour (red) only the sentences that were added or modified to address questions by the referees.

2) Please consider including comparisons to transfer learning methods, as advised by Reviewer #3. If feasible, comparisons to general transfer learning tools would be interesting and could strengthen the evidence for broad applicability. If these comparisons are not feasible, please explain why.

We explain in detail in response to Reviewer 3 why we believe these comparisons can be confusing. In short, a fair comparison of methods that were developed for different purposes is challenging, while comparing them on other datasets requires careful data curation and experimental setups. We explain these points in detail below.

Reviewer #2 (Recommendations for the authors):General QuestionsI will provide here some general questions that if addressed could, in my opinion, help improve the clarity of this article.1. I found the formulation aimed to disentangle the background from selected features really interesting. The way the learning is done is that both datasets are known a priori and then the learning happens sequentially. But what if the datasets where the two generic and specific features are mixed? How would the learning happen in this case? If DiffRBM cannot be used in this situation then it is ok, but it would be good to discuss it as a limitation.

We thank the reviewer for this interesting remark, we have added a sentence to comment on this point in the discussion (lines 532-535). In summary, the set up analyzed here is not designed to deal with these situations, but we think that some minimal modification of the framework would already be useful in this regard (e.g. allowing a kind of ‘parallel’ learning, where the learning of the background model not only informs the learning of the differential units, but also receives feedback from them).

2. The paper mentions that the learning happened in an HLA-specific approach. Although advantages and disadvantages are presented throughout the paper, it would be good to address this important distinction in the discussion.

We agree with the reviewer that this is a point worth covering again in the discussion, we have added a couple of sentences (lines 462-465).

3. Although I understand this article's main goal is to present a technical solution to specific phenomena in immunology, after reading the manuscript I felt that more specific direct applications should have been discussed. How can the scientific community, take this approach that has been extensively validated and compared and translate it into potential therapeutic or scientific applications?

We agree that this point deserves additional comments in the discussion. We have therefore added a short section on the potential applications of the method (vaccine design, cancer neoantigen discovery, TCR engineering, detection of immunoediting in cancer and immune evasion in viral evolution), see lines 505-508.

We have also expanded the discussion of the advantages and limitations of the method, in such a way that it becomes clearer in what cases it is convenient to use it (like the fact that the method is generative, sequence-based and parsimonious in terms of training data, see lines 468-497).

4. When describing contact prediction and the ranking of peptide positions, the result seemed quite encouraging, however, when one deals with such a small dataset (6-9 positions) then statistical significance becomes a relevant question. Can the authors also discuss this in the presentation of the results? This might help prove a more realistic representation of the predictive qualities of this side of DiffRBM.

We have derived an estimate of the statistical significance of our PPV values using a binomial test (as explained in the Methods, lines 784-793). The p-values are reported in the main text (lines 208-209, 396-397). At the same lines we have also emphasized more clearly in the text that the difference of our prediction with respect to the random expectation curve is statistically significant.

5. In Figure 3G the authors estimate the mutational cost of peptides compared to experimental measures. The correlation although significant seems low to me r=0.47. Can the authors comment if this is a limitation or if in fact, other methods cannot reach better correlations than the one presented using DiffRBM?

The correlation coefficient we find is not high but it is statistically significant (see the p-values provided in Figure 3 —figure supplement 1B). We look at this correlation not with the purpose of showing the predictive power per se of our presentation model (background RBM), but rather to support our hypothesis that these mutation costs, which are mutation costs measured by TCR avidity curves, reflect also an effect of disruption of presentation. In fact, these mutation costs are measured in terms of loss of TCR recognition, and therefore are expected to be predicted by a presentation model alone to a first approximation only (we should instead expect a higher correlation if one had measured the change in HLA-peptide binding). As an additional check, we have measured the same correlation coefficients using the predicted mutation costs by NetMHCpan4.1 (an established predictor of antigen presentation): also in this case, we find that their value is not particularly high (0.52 for TCR1, 0.45 for TCR2 and 0.66 for TCR3).

We have better clarified this point by rephrasing the sentences at lines 272-276.

Reviewer #3 (Recommendations for the authors):The authors propose a novel model architecture called diffRBM, which is based on the original RBM papers [Hinton, 2002, Hinton and Salakhutdinov, 2006], by adding separate background and differential units and a specific training procedure, as described in section 5.1, opening a possibility to use this model for the transfer of learning.To be interesting to the wide readership of eLife this paper should show novelty both in terms of the biology it describes and the(AI transfer of learning) tools. Especially as the tool's novelty is emphasized by the authors.While the paper is very long and detailed it is still a bit unclear to us why other than being novel in its AI methods it is better or more revealing than existing (also non-AI) methods in terms of our understanding of TCR immunology.We also have the following questions about the methodology:1. Since it is a novel transfer of learning method, we would like to see a general comparison of this method with other transfer of learning methods on standard benchmarks, not only on the immunological data. Authors claim in section 2 that "… it could be applied to any data that has some distinctive features compared to a much larger pool of data endowed with the baseline properties". We also believe that such a method could be useful to a broader community of researchers. Also, by performing a comparison on standard benchmarks, the novelty of diffRBM could be highlighted more clearly, since it will be certain whether this model consistently achieves state-of-the-art results or not.2. On the other hand, when applied to immunological data (that is described in the paper), we also would like to see more models from the classical and recent transfer of learning papers to be compared to. So if applicable, adding classical and recent transfer of learning models from the above links to the comparison would be beneficial.

We have decided not to carry out the comparisons to other data and to other transferlearning approaches suggested by the reviewer 3, because they might be not pertinent and largely outside the scope of our work for several reasons, summarizable as follows:

Additional comparisons would not be fair and consistent in terms of tasks, and would not serve to strengthen our main message;Additional comparisons do not fit the stated purpose of our work;We do already have comparisons to similar methods when it is feasible, consistent in terms of type of task, and relevant;Additional comparisons require an amount of work that would justify separate papers.

We now elaborate more on each of these reasons.

Additional comparisons would not be fair and consistent in terms of tasks, and would not serve to strengthen our main message. Many machine learning papers on transfer learning and domain adaptation focus on obtaining domain-invariant representations and portability of methods across domains, which are not the tasks we are interested in here. Since these methods were developed for other purposes, the comparisons we would carry out would neither be fair, nor consistent in terms of tasks for which the different methods are optimized.

Our main message is that by learning the statistical differences between ‘background' and biologically selected data we can obtain insight into selected biological features, in particular the enriched molecular motifs in immunogenic antigens and epitope-specific receptors. The model parameters learnt in such a way as to capture such statistical differences allow us to extract biologically interpretable predictions, like contact sites, mutation costs in terms of TCR reactivity, patterns of enrichment in certain biological properties. The typical transfer-learning task of designing domain-invariant representations goes in a different direction, thus, even if we compared to other methods from the field of domain adaptation, the result would not help us illustrate these points.

Additional comparisons do not fit the stated purpose of our work. This work is not meant to be a purely machine learning paper, our focus on the inspection and extraction of biologically interpretable features in an immunological context is pivotal. We are not putting forward claims on diffRBM being a better transfer-learning method than other ones in general, and it is not our aim to compete with them. Rather, the novelty of our approach relies on the idea of using transfer-learning for these two specific problems (modeling antigen immunogenicity and T-cell receptor specificity).We do already have comparisons to similar methods when it is feasible, consistent in terms of type of task, and relevant. In terms of model performance, we already have the comparison to the most established predictors in both fields (prediction of antigen immunogenicity and TCR specificity). Importantly, these comparisons already include the existing transfer-learning methods in T-cell receptor modeling: one of them is SONIA [Elhanati et al., 2014, Sethna et al., 2020], to which we compare diffRBM explicitly by training and testing it on our same dataset. Another one is TITAN [Weber et al., 2021], that exploits a pre-training on general protein-protein interactions data to build a model of TCR-epitope binding. Since in a recent benchmark of different methods [Meysman et al., 2022] TITAN performed slightly worse than TCRex and NetTCR , we decided to include explicitly the comparison only with TCRex and NetTCR (the most performant methods to date). Given that diffRBM can perform as well as TCRex and NetTCR at the task of distinguishing specific from generic receptors from the background dataset, the results in [Meysman et al., 2022] suggest that diffRBM would compare favorably also to the other methods tested including TITAN. Hence our work uses previously done benchmarks to identify the best methods with which it is sensible to compare. In connection to modeling antigen immunogenicity, we are not aware of transfer-learning methods. One of the methods we consider for comparison (the IEDB tool, based on [Calis et al., 2013]) is based on the similar idea of measuring the enrichment in amino acid usage between immunogenic and non-immunogenic peptides, but with the simple assumption of position-independence of the main biophysical properties underlying immunogenicity.

Next, we wanted to emphasize that the approach is parsimonious in terms of training data, an aspect that is particularly crucial since data annotated by the information on immunogenicity or epitope specificity are scarce and the risk of overfitting is high. In this regard, other comparisons are important, i.e. comparisons to linear models with fewer parameters (diffRBM linear, PWM, SONIA, linear classifier), and we carry them out systematically (Figures 2, 4, 5, 6, Figure 4 —figure supplement 1, Figure 6 —figure supplement 1). These comparisons illustrate that diffRBM is a more advanced yet parsimonious choice, which is able to capture nonlinearities at the price of a limited amount of added parameters, keeping us away from overfitting but at the same time giving improved performance (as stressed at lines 474-477).

Additional comparisons require an amount of work that would justify separate papers. It is true that we mention that the diffRBM approach could be eventually applied to data that possess some distinctive, enriched set of features compared to larger pools of data endowed with the baseline properties. In particular, specific types of data we have in mind are RNA sequences selected through Selex protocols and protein sequences produced by in vitro directed evolution. We mentioned, however, such applications with different biological data only as future directions because they would require an amount of additional work that is even superior to the one put into this manuscript. Indeed, the application to new data requires a substantial work of model selection and testing (as done here), but also to design and test pipelines of data pre-processing specific to the new data. The steps of pre-processing and embedding required for the biological data we have analyzed here are quite ad hoc and demanding per se: for these steps we are actually building on the efforts of two separate papers, one on the application of RBMs to model T-cell receptors [Bravi et al., 2021a] and one on the application of RBMs to model antigen presentation [Bravi et al., 2021b]. This is the same reason why also performing comparisons of diffRBM to transfer-learning approaches implemented through different machine machine learning architectures, for instance from the papers kindly suggested by the reviewer, would require an amount of work that would justify a separate paper. Other machine learning architectures simply can’t be used off-the-shelf on these sequence data. As we have highlighted in the text for the comparisons we carry out, re-training other methods on the same data is key to a fair comparison, and, in addition to re-training, designing and implementing the steps of data pre-processing and embedding for other model architectures would require an amount of work that goes beyond the scope of a simple comparison of transfer-learning ideas.

To clarify all these points, we have made the following changes to the text:

In the introduction, we have clarified that the main focus is on the inspection and extraction of biologically interpretable features through the learnt parameters in terms of these properties (lines 87-89). We have expanded the paragraph where we mention other transfer-learning approaches in the context of modelling T-cell receptor specificity, to clarify how diffRBM is positioned compared to these approaches, in terms of architecture, training, and related advantages (lines 105-114).In the discussion, we have added a paragraph on the difference of diffRBM with typical tasks for which transfer-learning is used (portability across domains), thus we stress diffRBM is not meant to compete with this type of approach (lines 498-504).We have left the reference to additional applications with new data (such as data from SELEX protocols) only in the discussion about future extensions. We clarify that these applications are meant as future work since they will require the design of pipelines of data pre-processing specific to the new data, and possibly other adjustments of the model architecture and the learning strategy to accommodate their specificities (lines 529-532).